# Optimising genomic approaches for identifying vancomycin-resistant *Enterococcus faecium* transmission in healthcare settings

Charlie Higgs[1], Norelle L. Sherry [1,2,3], Torsten Seemann [1,2], Kristy Horan[2], Hasini Walpola[2], Paul Kinsella [4], Katherine Bond[4], Deborah A. Williamson[2,4,5], Caroline Marshall[6], Jason C. Kwong[1,3], M. Lindsay Grayson[3,7], Timothy P. Stinear [1,8], Claire L. Gorrie[1,2,8] & Benjamin P. Howden [1,2,3,8 ✉]

Vancomycin-resistant *Enterococcus faecium* (VREfm) is a major nosocomial pathogen. Identifying VREfm transmission dynamics permits targeted interventions, and while genomics is increasingly being utilised, methods are not yet standardised or optimised for accuracy. We aimed to develop a standardized genomic method for identifying putative VREfm transmission links. Using comprehensive genomic and epidemiological data from a cohort of 308 VREfm infection or colonization cases, we compared multiple approaches for quantifying genetic relatedness. We showed that clustering by core genome multilocus sequence type (cgMLST) was more informative of population structure than traditional MLST. Pairwise genome comparisons using split k-mer analysis (SKA) provided the high-level resolution needed to infer patient-to-patient transmission. The more common mapping to a reference genome was not sufficiently discriminatory, defining more than three times more genomic transmission events than SKA (3729 compared to 1079 events). Here, we show a standardized genomic framework for inferring VREfm transmission that can be the basis for global deployment of VREfm genomics into routine outbreak detection and investigation.

[1] Department of Microbiology & Immunology, The Peter Doherty Institute for Infection and Immunity, The University of Melbourne, Melbourne, VIC, Australia. [2] Microbiological Diagnostic Unit Public Health Laboratory, Department of Microbiology & Immunology, The Peter Doherty Institute for Infection and Immunity, The University of Melbourne, Melbourne, VIC, Australia. [3] Department of Infectious Diseases, Austin Health, Melbourne, VIC, Australia. [4] Department of Microbiology, Royal Melbourne Hospital, Melbourne, VIC, Australia. [5] Victorian Infectious Diseases Reference Laboratory, Royal Melbourne Hospital, The Peter Doherty Institute for Infection and Immunity, Melbourne, VIC, Australia. [6] Victorian Infectious Diseases Service, The Peter Doherty Institute for Infection and Immunity, Melbourne, VIC, Australia. [7] Department of Medicine, Austin Health, The University of Melbourne, Heidelberg, VIC, Australia. [8] These authors jointly supervised this work: Timothy P. Stinear, Claire L. Gorrie, Benjamin P. Howden. ✉email: bhowden@unimelb.edu.au

*E*nterococcus faecium comprises part of the normal gastro-intestinal tract flora in humans, but in recent years, has become an increasingly important healthcare-associated pathogen, causing a range of infections, including bloodstream and urinary tract infections[1–3]. *Enterococcus faecium* is particularly difficult to treat due to intrinsic resistance to a range of antibiotics, and although vancomycin was once the treatment of choice for *E. faecium*, vancomycin-resistant *E. faecium* is now common in many countries[4]. Consequently, the World Health Organisation (WHO) listed vancomycin-resistant *Enterococcus faecium* (VREfm) as a "high priority" in its recent list of priority bacterial pathogens due to the limited treatment options available[5].

Whole-genome sequencing (WGS) allows high-resolution investigation of pathogen outbreaks and transmission networks, facilitating accurate targeting of infection control interventions and consequently reducing the number of infections[6]. Conventional genomic approaches for determining genetic relatedness and transmission links include multilocus sequence type (MLST) and single nucleotide polymorphism (SNP) distances, generated from core genome alignments usually using recombination masking. However, these methods are not optimal for highly recombinogenic and very genetically diverse species, such as *E. faecium*, and may result in isolates being labelled as more closely/less closely related than they actually are[7]. More recently, alternative approaches such as core genome MLST (cgMLST) and kmer-based methods (genomic data broken down into smaller pieces of known size k) have been developed to deal with some of the pitfalls of the conventional approaches. In all approaches, genomic data are combined with epidemiological information for a more accurate determination of transmission links and outbreak status.

Multiple studies have demonstrated the utility of these methods for VREfm, especially when there are complex transmission networks that involve multiple wards or when hospitals have a high VREfm burden[8–10]. Despite this proven utility, there is yet to be standardisation in the methods and thresholds used to identify outbreaks and transmission links. The lack of standardisation is a key factor identified by the WHO limiting the use of whole-genome sequencing in routine surveillance of antimicrobial resistance[11]. Multiple thresholds have been proposed for pairwise SNP distances to identify putative transmission links in *E. faecium* based on within-patient isolate diversity (ranging from 6 to 17 SNPs)[10,12,13]. However, these thresholds are not accompanied by methods to standardise the preceding steps in the analyses. A recent study from our group demonstrated the large variation in SNP distances caused by changes in analysis approach, such as reference choice, masking for recombination and subsampling of datasets[7]. For analysis of VREfm, this study recommended using a closely related reference genome and not masking for prophage/recombination regions to ensure consistent and accurate pairwise SNP distances.

Current genomic studies on transmission and outbreak analysis are primarily focussed on retrospective analysis that only occurs once an outbreak is suspected or has been resolved. Laboratory and bioinformatic workflows that identify potential transmission events in closer to real time are still being developed. One potential system was described by Brown et al., in which methicillin-resistant *Staphylococcus aureus* (MRSA) were sequenced and the data interpreted using an automated pipeline[14]. Promisingly, the data showed that even with current technology, the limiting step in real-time outbreak detection using WGS is the data analysis step and not the turnaround time of sequencing. For a bioinformatic analysis of genomic transmission to be implemented as standard practice it needs to: (i) be stable over time as additional isolates are added to the analysis; (ii) be standardised to allow for comparison across sites or hospitals; (iii) be computationally efficient; and (iv) allow for automation and require minimal intervention and interpretation.

The aim of this study was to determine the best genomic approach for identifying putative VREfm transmission links in as close to real-time as possible and develop an analysis pipeline that could be implemented as standard practice. Using WGS and comprehensive patient location data, we compared genomic variant calling methods and pairwise SNP distance thresholds for transmission inference to develop a method that best fulfilled these requirements.

Here, we show the superiority of using cgMLST clustering as an alternative to MLST to group genetically similar *E. faecium* isolates and using split kmer analysis (SKA) to identify putative transmission clusters based on genomic similarity. In addition, we demonstrate how the use of these methods in a health care scenario provide accurate information on transmission events and allow for the specific targeting of infection control resources.

## Results

**Isolate clusters identified with cgMLST more accurately represent the population structure and isolate relatedness when compared to traditional MLST.** MLST has traditionally been used to broadly group genetically similar isolates together within a species, including when identifying isolates worth investigating for possible transmission. We aimed to compare groups identified using traditional MLST and those identified using cgMLST to see which best matched the population structure.

We first constructed a phylogeny in order to establish the population structure of the local *vanA* VRE population (Fig. 1a). This revealed that three of the most common STs (ST203, ST80 and ST1421) were interspersed throughout the tree and the range of SNP distances of within-ST pairs significantly overlapped with distances of between ST pairs for all major ST backgrounds (Supplementary Fig. 1).

cgMLST was tested as an alternative approach. The number of cgMLST allele differences was calculated between all isolate pairs (total number of genes in scheme $n = 1,423$), both within and between STs. The distribution of cgMLST pairwise allelic difference values across the species shows a clear peak at the lower values that only contains isolate pairs from the same ST (Fig. 1b). It is not until cgMLST pairwise allelic differences of $\geq 42$ that the number of between-ST isolate pairs observed is greater than the number of within-ST isolate pairs. The smallest cgMLST allelic difference value of a pair of isolates arising from different STs was 26. The bottom of the first trough also occurs close to this value at 25 cgMLST allelic differences (Fig. 1b). This is true when looking at the species level as well as individual STs (Supplementary Fig. 2). The value of $\leq 25$ allelic differences was then used as a threshold to generate cgMLST clusters for all VRE isolates ($n = 346$) through a single-linkage cluster approach. This meant no between-ST isolate pairs were clustered together while still maximising the number of within-ST isolate pairs.

Using the traditional MLST scheme, 18 STs were identified, of which eight were singletons, containing only one isolate each. In comparison, when the cgMLST approach was used with a pairwise allelic difference clustering threshold of $\leq 25$, 63 cgMLST clusters were identified including 40 singletons (Supplementary Fig. 3). When considering the four most common local *vanA* VRE STs as key examples (ST80, ST203, ST1421 and ST1424), this allelic difference threshold also separated isolate pairs of the same ST in a manner that better corresponded with the population structure (Fig. 1a). The range of pairwise SNP distances, drawn from the core alignment used to generate the species phylogeny, of within-cgMLST clusters compared to within-STs was also dramatically reduced (Supplementary Fig. 1).

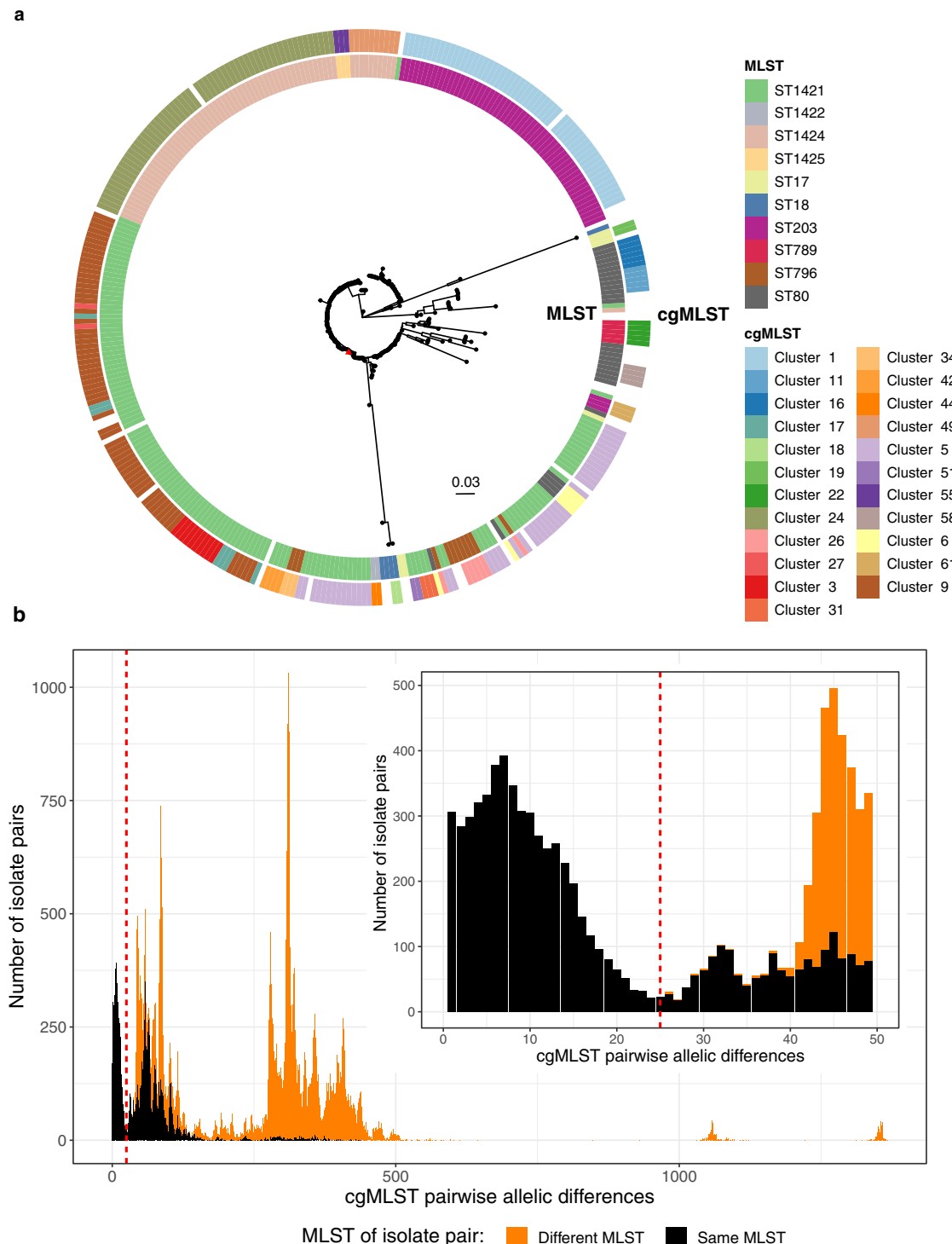

**Fig. 1 Comparison of genomic typing methods for E faecium. a** Midpoint-rooted maximum-likelihood phylogenetic tree of *E. faecium* isolates. Tree includes all isolates used in this study not including genetic outliers (*n* = 343). The reference genome is identified by the red triangle tip and is sequence type (ST) 1421. Singleton STs and core genome multilocus sequence type (cgMLST) clusters with only one isolate are shown in white. **b** Stacked histogram of cgMLST allelic differences between all *E. faecium* isolate pairs (*n* = 346 isolates). The total number of genes in the scheme is 1423. The red dashed line represents a pairwise allelic difference threshold of 25. The inset graph shows the same data set but has been restricted to show only cgMLST pairwise allelic difference of ≤ 50.

These results suggest that isolate clusters identified with cgMLST, and ≤25 allelic difference single-linkage clustering, more accurately represent the population structure and isolate relatedness when compared to traditional MLST and should be used instead.

**For very closely related isolate pairs, direct pairwise comparison tools should be used**. After genetically similar isolates have been broadly grouped together, i.e. with MLST or cgMLST clustering, a finer-scale comparison is needed to identify possible transmission events. Traditionally, this involves constructing core

genome alignments and then calculating a pairwise SNP distance between all isolate pairs. Numerous fine-scale genetic comparison approaches were compared in order to determine which correlated best to pairwise comparison using *de novo* references (PCDR), the gold standard comparison method.

The isolate pairwise distances calculated using the multi-locus sequence type core alignments (MLSTCA), core genome MLST core alignments (cgMLSTCA) and core genome MLST with cluster references core alignments (cgMLSTCRCA) approaches did not correlate well with PCDR distances. (i.e., pairs that are identified as closely related by one method are not always identified as closely related by PCDR), however, this was not true of split kmer analysis (SKA) which consistently correlated well with PCDR ($R^2 > 0.83$) (Supplementary Fig. 4a).

This was true even when considering isolate pairs that are closely related (PCDR pairwise SNP distance ≤ 50), the $R^2$ values comparing PCDR distances and those derived from MLSTCA, cgMLSTCA were again consistently low, particularly those comparing the ST80, ST1424 and ST1421 backgrounds ($R^2 < 0.24$) (Supplementary Fig. 4b). cgMLSTCRCA, with references more closely related to the sample group, correlated better with PCDR values than MLSTCA and cgMLSTCA across all ST backgrounds but $R^2$ values never reached more than 0.69. SKA distances were strongly correlated ($R^2 > 0.74$) across all ST backgrounds, representing the strong correlation between the two methods.

These data clearly show that for very closely related isolate pairs, when using PCDR as the gold standard, core genome alignments of cgMLST cluster isolates provide an inaccurate measure of relatedness and direct pairwise comparison tools should be used instead. The pairwise distance data from MLSTCA was not analysed further due to the consistently poor $R^2$ values. MLSTCA had the smallest average core alignment length as a percentage of reference (66%), while cgMLSTCA and cgMLSTCRCA were comparable with an average of approximately 80% (Supplementary Data 1 and 2).

**Pairwise genetic comparison methods produce the most similar putative transmission clusters.** After quantifying the genetic distance between isolates, a threshold was set based on intrapatient diversity in order to identify genomic putative transmission links (isolate pairs below the threshold). This threshold differed between genomic methods with cgMLSTCRCA and SKA having the most similar thresholds (≤6 SNPs and ≤7 SNPs respectively) and PCDR having a threshold of ≤10 SNPs and cgMLSTCA ≤44 SNPs (Supplementary Fig. 5). These thresholds were suggestive of 3704, 1363, 1023 and 1054 genomic putative transmission links for cgMLSTCA, cgMLSTCRCA, PCDR and SKA, respectively (Supplementary Fig. 6).

Single linkage clustering was then used to group isolates based on the above SNP thresholds to determine genomic putative transmission clusters. The clustering patterns of cgMLSTCA and did not match cgMLSTCRCA (Fig. 2 and Supplementary Fig. 7), with cgMLSTCA putative transmission clusters containing more isolates than all other methods across all ST backgrounds. Clusters identified through cgMLSTCRCA better-reflected PCDR clusters, however, there were still a number of inconsistencies with cgMLSTCRCA grouping isolates together that were separated by PCDR (particularly in ST1421 and ST1424 backgrounds). The clustering patterns of SKA and PCDR were very similar across all ST backgrounds.

Of the PCDR singleton clusters (i.e., single isolates not genomically linked to any other isolate $n = 48$), five were linked to other isolates by SKA and 13 were linked to other isolates by cgMLSTCRCA. Of the SKA ($n = 44$) and cgMLSTCRCA ($n = 42$)

clusters with one isolate, one and seven respectively were genomically linked to another isolate by PCDR. Of the PCDR clusters containing more than one isolate, cgMLSTCRCA split 7 while SKA split only two (Supplementary Fig. 8). Data used to conduct the above analyses can be found in Supplementary Data 3.

**cgMLSTCRCA, PCDR and SKA all had a similarly high level of genomic putative transmission links that were supported by epidemiological evidence.** The epidemiological relationship between isolate pairs were compared to genomic putative transmission links to determine the extent they were supported by epidemiological evidence. In addition to identifying a much greater number of genomic putative transmission links, cgMLSTCA also had the greatest percentage of links that were not supported by epidemiological evidence (46%, more than double any other genomic method) (Fig. 3 and Supplementary Fig. 6). A similarly high proportion of genomic putative transmission links were supported by epidemiological evidence (probable and possible transmission likelihood) for cgMLSTCRCA (80%), PCDR (83%), SKA (83%).

**Use of optimal genomic analysis pipeline provides valuable insights into VREfm transmission.** Based on the results of the previous analysis and the method comparison in Fig. 3, an optimal genomic analysis pipeline was determined. This pipeline is outlined in Fig. 4. Using the proposed pipeline on a contained single hospital case study identified 26 patients involved in transmission event across six cgMLST clusters and nine genomic transmission clusters (Fig. 5). Comparison to epidemiological data revealed that two of these clusters only contained isolates from the same patient. Of the remaining seven genomic transmission clusters, five contained a combination of both screening and clinical isolates. When comparing the genomic transmission clusters to epidemiological data, 22 of the 71 genomic links (30%) were also supported by epidemiological connections of the same ward at the same time. Of the remaining 49 genomic links, 5 were isolates from the same patient and 16 (23%) had been on the same ward ≤60 days apart. Examination of the wards involved in the epidemiological links revealed that each genomic cluster contained one or two wards that were responsible for all same ward at the same time links (Supplementary Fig. 9). A single ward (Ward 5) had undergone enhanced VRE screening, and although transmission events involving this ward were identified, clusters also included patients that had not spent time on this ward (Supplementary Fig. 9). The exact pairwise distances of all isolates below the same patient threshold can be found in Supplementary Data 4.

**Discussion**
Whilst MLST groups broadly correspond with phylogenetic population structure for many species, the current MLST scheme for *E. faecium* does not accurately reflect its population structure, indicating that it may not be the most appropriate method for identifying related isolate groups for further transmission analyses (Fig. 1). Clusters defined by cgMLST using an allelic-difference threshold provide a much more accurate reflection of population structure, with cgMLST clusters matching phylogenetic clusters (Fig. 1). When categorising isolate groups, cgMLST incorporates significantly more genomic content than MLST but still retains many important features of the MLST scheme such as not being dataset-dependent and being able to compare between datasets in a standardised manner. We therefore suggest that cgMLST clustering should be used instead of MLST to initially identify genetically similar isolates worth investigating for

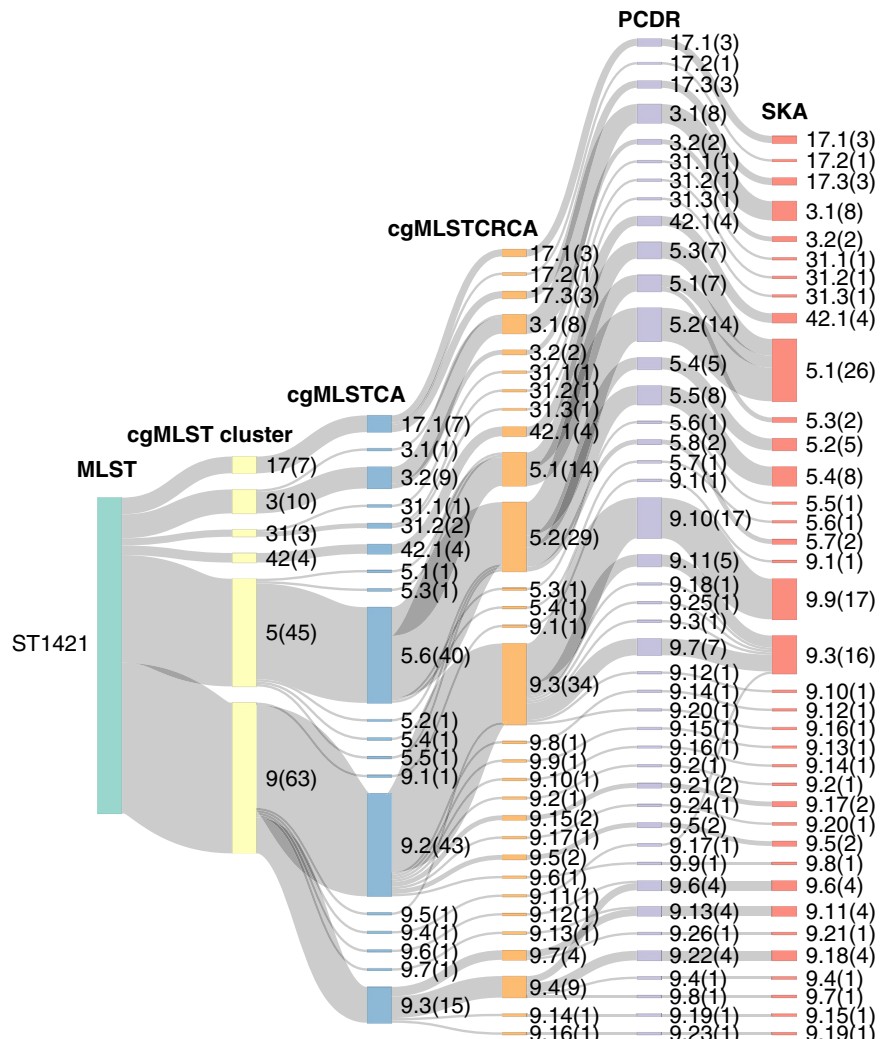

**Fig. 2 Relationship between methods for determining genomic clusters for *E. faecium* isolates from the major multilocus sequence type (ST1421 *n* = 134).** Core genome multilocus sequence type (cgMLST) clusters are based on single-linkage clustering using a pairwise allelic difference threshold of ≤25 alleles. cgMLST core alignment (cgMLSTCA), cgMLST with cluster reference core alignments (cgMLSTCRCA), pairwise comparison using *de novo* references (PCDR) and split kmer analysis (SKA) clusters are based on a pairwise single nucleotide polymorphism (SNP) distance threshold determined based on intrapatient SNP diversity (cgMLSTCA: ≤44 SNPs, cgMLSTCRCA: ≤6 SNPs, PCDR: ≤10 SNPs, SKA: ≤7 SNPs) and are generated using single linkage clustering. The size of the nodes represents the number of isolates in each of the clusters and is relative for each multilocus sequence type (MLST) and the number of isolates in each cluster is displayed in brackets. PCDR was used as the gold standard method and so the genomic clustering pattern of the other methods should be as close as possible to PCDR.

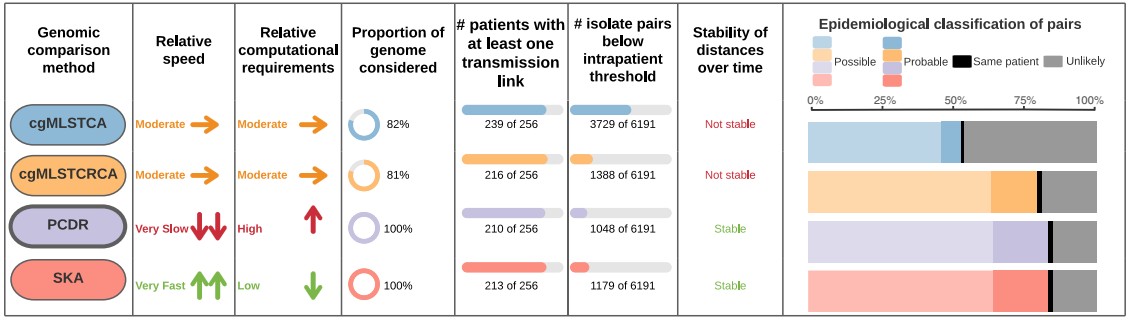

**Fig. 3 Comparison of the major attributes of each of the methods used to determine genetic similarity.** The number of isolate pairs below the intrapatient threshold was determined individually for each of the genomic methods and the epidemiological classification of isolate pairs only occurred for pairs below the threshold. The epidemiological classification process of the isolate pairs is detailed in the Supplementary Fig. 10. Pairwise comparison using *de novo* references (PCDR) was used as the gold standard genomic comparison method.

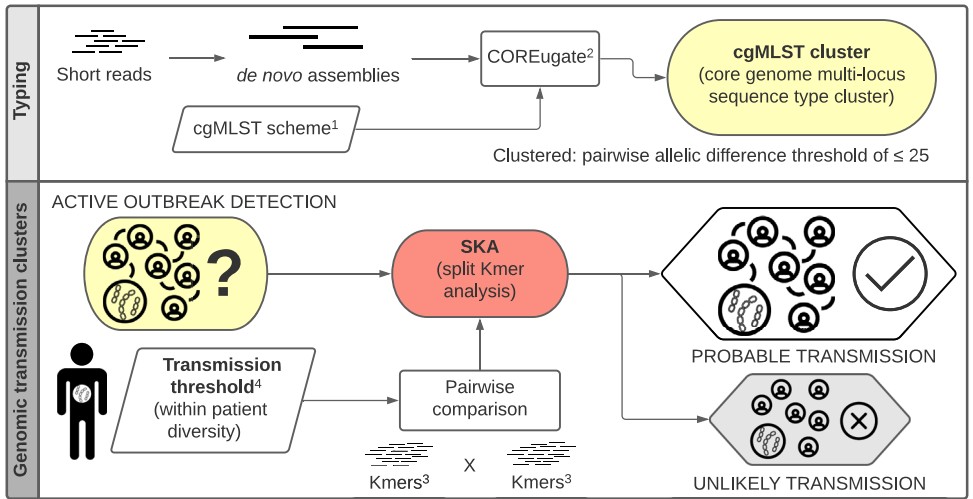

**Fig. 4 Proposed genomic analysis pipeline for identifying potential vancomycin-resistant *Enterococcus faecium* transmission.** Outlined is each step in the analysis pipeline, the software used and the required whole genome sequencing data. [1] core genome multiloicus sequence type (cgMLST) allele scheme by de Been et al. [2] The COREugate pipeline was used to assign the allelic profiles and build a matrix of the pairwise allelic differences. [3] Split kmers were generated from short read data at $k = 15$. [4] The threshold for determining transmission events was determined based on within patient diversity. For this data set and split kmer analysis (SKA) this was found to be 7 single nucleotide polymorphisms (SNPs).

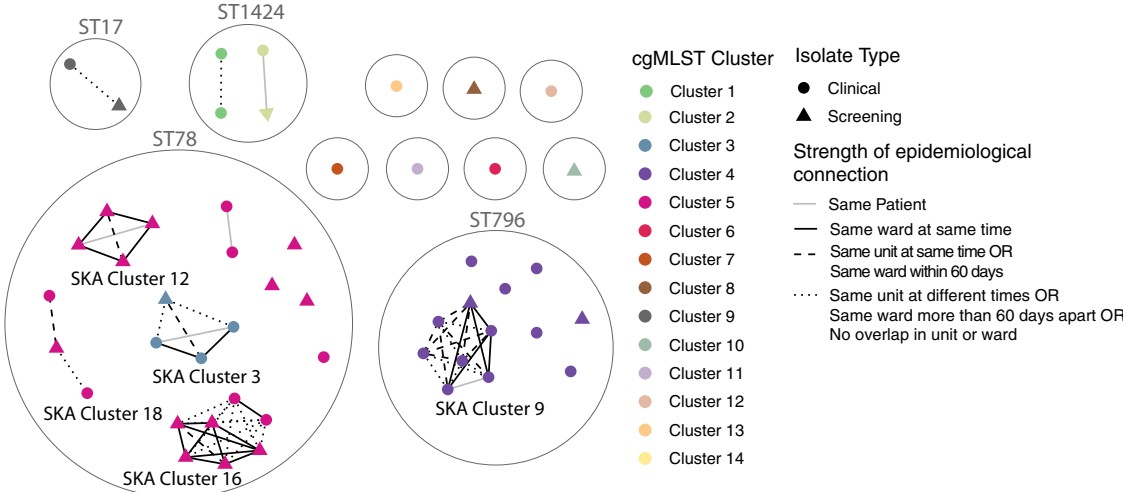

**Fig. 5 Transmission networks identified in the hospital case study.** Network diagram of genomics transmission links of all isolates involved in the hospital case study. Each node represents one case and are coloured by their respective core genome multilocus sequence type (cgMLST) cluster. The shape of each node identifies if the isolate was collected as patient screening or as a clinical sample. Nodes are grouped in grey circles based on their sequence type (ST) (only STs with more than one isolate have been labelled). Lines between nodes are present if the split kmer analysis (SKA) pairwise distance between isolates is ≤7 single nucleotide polymorphisms (SNPs) and the line type represents the strength of epidemiological link between the isolates. SKA clusters with more than 3 isolates have been labelled.

transmission links, as the cgMLST clusters are more consistent in the level of intra- and inter- cluster genomic diversity compared to MLST.

When using genomics for VRE transmission surveillance it is essential that analysis methods are reliable and valuable infection control resources are not wasted by investigating cases that are not truly linked or missing cases that are. Therefore, we aimed to identify the best analysis approach for identifying genomic putative transmission links. We tested multiple core alignment methods, each time changing either the clustering of samples or the relatedness of the reference genome. These approaches are similar to those that are conventionally used when performing genomic transmission analyses. We also tested two pairwise comparison methods in order to determine how well these

correlated with the core alignment methods and if all methods labelled the same isolate pairs as closely related (i.e., genomic putative transmission link) (Fig. 6).

It was hypothesised that cgMLSTCA (cgMLST clusters, MLST reference) pairwise distances would more closely reflect the PCDR (direct pairwise alignment) distances compared to MLSTCA (MLST clusters, MLST reference) distances. Although cgMLSTCA (cgMLST clusters, MLST reference) pairwise distances better correlated with PCDR (direct pairwise alignment) distances compared to MLSTCA (higher $R^2$ values), reflective of this change in clustering method, cgMLSTCRCA (cgMLST clusters, cgMLST reference) distances had an even stronger correlation to PCDR (direct pairwise alignment) distances (higher $R^2$ values), reflective of the more closely related reference genomes.

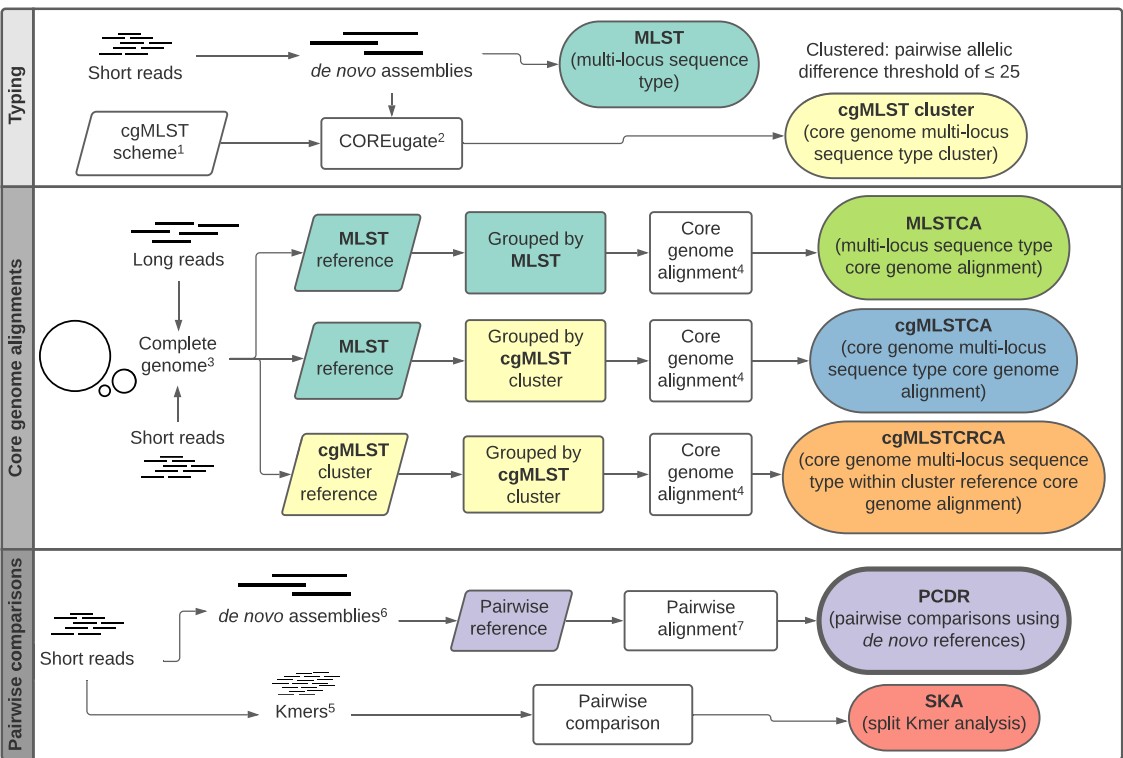

**Fig. 6 Methods for investigating the genomic diversity of vancomycin-resistant *E. faecium*.** This flow chart provides an overview of all the genetic comparison approaches used in the study. It has been split into three sections based on the type of analysis that each approach uses: typing, core genome alignments and pairwise comparisons. Pairwise comparison using *de novo* references (PCDR) was used as the gold standard method for determining genomic diversity. [1] core genome multilocus sequence type (cgMLST) allele scheme by de Been et al. [2] The COREugate pipeline was used to assign the allelic profiles and build a matrix of the pairwise allelic differences. [3] Complete genomes were assembled using Unicycler/Canu (multilocus sequence type (MLST) references) or Trycycler (cgMLST references). [4] Core genome alignments were performed using short-read data and snippy. Pairwise distances were then generated from the core alignment. [5] Split kmers were generated from short read data at $k = 15$. [6] *de novo* assemblies were performed using SKESA. [7] Individual snippy alignments were performed using the *de novo* assemblies and short read data on each pair. Any self-single nucleotide polymorphisms identified from mapping self reads to the *de novo* assembly were removed. The mean of the two reciprocal alignments was then used as the pairwise distance.

This is consistent with a recent study by Gorrie et al. that detailed the large effects that reference choice and sample diversity can have on core alignments and showed that there were significant differences in the distribution of pairwise distances depending on how closely related the reference genome was[7]. Although generating reference genomes for each cgMLST cluster is possible, it is an additional step that adds to the time and cost of the analysis pipeline. As such, a method that does not require a within cgMLST cluster *de novo* reference but can still accurately replicate PCDR (direct pairwise alignment) results without the intense computational requirements is preferred. Across all the major ST backgrounds, SKA (direct pairwise kmer) and PCDR (direct pairwise alignment) pairwise distances were consistently well correlated, even when comparing isolates that were very closely related. SKA (direct pairwise kmer) is significantly faster than PCDR (direct pairwise alignment) and requires much fewer computational resources, both in terms of processing and memory. Unlike cgMLSTCA (cgMLST clusters, MLST reference), SKA (direct pairwise kmer) does not require *de novo* reference genomes and the pairwise distances are not affected by changes in sample diversity, making it an ideal alternative.

When translating genomic putative transmission links into putative transmission clusters, via single-linkage clustering, there were a similar number of inconsistencies in cgMLSTCRCA (cgMLST clusters, cgMLST reference) and SKA (direct pairwise kmer) clusters compared to PCDR (direct pairwise alignment). Minor errors in genomic putative transmission linkage were classified as isolates identified as being involved in transmission by the respective genomic method but not by the gold standard (PCDR). SKA (direct pairwise kmer) had five minor errors and cgMLSTCRCA (cgMLST clusters, cgMLST reference) had 13. Major errors in genomic putative transmission linkage were classified as isolates identified as being involved in transmission by the gold standard genomic method (PCDR) but not by the respective genomic method. SKA (direct pairwise kmer) had one major error and cgMLSTCRCA (cgMLST clusters, cgMLST reference) had seven. However, the practical benefits of SKA (fast, not computationally intensive, and pairwise SNP distances stable over time) and the extra sequencing requirements for cgMLST cluster references in cgMLSTCRCA (cgMLST clusters, cgMLST reference) mean that SKA (direct pairwise kmer) is likely to be a much more desirable method in a real-world scenario.

The pairwise SNP distance genomic transmission thresholds derived from intra-patient diversity for cgMLSTCRCA (cgMLST clusters, cgMLST reference), PCDR (direct pairwise alignment) and SKA (direct pairwise kmer) were similar in magnitude to those previously described[10,12,13]. However, we have shown that the methods used in these studies (ST-based core alignments and in some cases species core alignments) do not accurately reflect the degree of relatedness between all isolate pairs and consequently can result in mislabelling isolates as involved or not involved in transmission. This can have large implications in the hospital environment where the use of finite infection control resources needs to be as targeted as possible. This phenomenon

was observed in the cgMLSTCA transmission threshold, which was considerably higher than the other methods and subsequently classified a much higher number of isolates as involved in transmission. Additionally, sequencing multiple colonies from individual cultures would have provided a more accurate assessment of the level of intra-patient genomic diversity. Although this was not performed in our study, we used data from Gouliouris et al.[13], who sequenced extensively from the same patient, along with our preferred genomic comparison method SKA and found an equivalent threshold of ≤5 SNPs. Although slightly lower than our SKA threshold of ≤7 SNPs, we felt it appropriate for the threshold to remain at ≤7 SNPs to ensure no potential transmission pairs were missed.

In summation, to identify genomic putative transmission clusters we propose using cgMLST clustering to broadly group genetically similar isolates and then split kmer analysis on the isolates within these cgMLST clusters (SKA) to identify putative transmission links. This proposed pipeline is outlined in Fig. 4. SKA fulfills all the proposed requirements of an implementable analysis pipeline: the pairwise differences generated are stable over time as additional isolates are added to the analysis; the analysis can be easily standardised with SNP transmission thresholds; it requires minimal computational resources; and can be easily automated (Fig. 3). These attributes mean that the method we have presented can be used prospectively with data interpretation happening immediately after sequencing and quality control has finished. In addition, the standardised nature of this approach would allow the prevalence of VRE transmission to be compared between hospitals, making it easier to understand the effects of different screening, cleaning, and isolation practices. Although setting a hard SNP threshold for identifying genomic transmission events leads to some ambiguity in the status of isolate pairs with distances very close to the threshold, it does mean that such a method can be more easily applied without requiring extensive bioinformatic expertise. The systematic approach used in this study to identify the optimal analysis method for identifying putative transmission links could be applied to the investigation and tracking of other major AMR pathogens. The ability of this approach to appropriately compensate for large differences in the genomic diversity of VRE STs is further evidence of its applicability to other species.

Implementation of this proposed method into routine use relies on it having added clinical value which was demonstrated in the hospital case study. Use of cgMLST was able to differentiate the ST78 isolates into two distinct clusters (cgMLST cluster 3 and cgMLST cluster 5). These well-defined transmission clusters allowed for easy identification of high-risk wards which may inform targeted interventions. The use of SKA to identify genomic clusters allowed the differentiation of singular cgMLST clusters into multiple distinct genomic transmission clusters (ST78) (Fig. 5), of which one cluster (cluster 3) was associated with completely different wards. This case study also highlighted the importance of including both clinical and screening isolates in the analysis as 5 SKA genomic transmission clusters contained isolates from screening and clinical samples. In addition, when comparing the genomic clusters to epidemiological data, clear wards of interest could be defined based on the predominant ward in the cluster where patients overlapped in space and time, allowing for the direct targeting of infection control resources. This case study further reinforces the importance of a coordinated and hospital-wide transmission detection approach and demonstrated that the approach we have defined can be applied effectively in a real-world setting to identify transmission events and clusters.

This study relies on the isolate sequences from the Controlling Superbugs study, and its limitations. Transmission of VREfm in

hospitals is inherently difficult to study, depending heavily on the screening strategies employed at each hospital, and the veracity of epidemiologic data used to define likely transmission (in combination with genomic data in this case). In the Controlling Superbugs study, sampling strategies differed between hospitals, with some hospitals sampling patients more extensively than others and hence making it more likely that putative transmission links are identified. Although the ward move data is highly detailed, it can contain errors and may miss other patient contacts. These errors may explain some of the genomic putative transmission links that were not supported by epidemiological data. However, the purpose of this study was focussed on exploring the differences in genomic methods rather than quantifying the exact number of the transmission events that have occurred and their directionality and source. Hence these limitations have minimal impact on the results in this case. Differences in genomic relatedness caused by variant calling software (snippy and SKA) and input parameters may also have resulted in the erroneous classification of isolates as linked. However, snippy has been shown to be among the best bacterial SNP callers in a comparison by Bush et al. and the purpose of this study was to explore the key issues with using core alignments to identify putative transmission links rather than the effect of changing the variant caller[15]. Hence, we chose to use snippy as the variant caller for all alignment methods. Using single linkage clustering to cluster cgMLST may result in the joining of clusters over time as isolates continue to be added to the analysis, however, we believe the extra detail provided by this method outweighs this limitation and can be managed with appropriate nomenclature and education of the recipients of these data. Although a relatively small number of intra-patient pairs were used to define the threshold for genomic transmission links making it more susceptible to outliers, the threshold is consistent with previous studies and we therefore suggest the threshold is appropriate[10,12,13].

In conclusion, we demonstrate the benefits of using cgMLST clustering as an alternative to MLST to cluster genetically similar *E. faecium* isolates. We have then explored how these clusters can be analysed to identify putative transmission links based on genomic similarity and suggest that the optimal method is SKA. These data will inform the future translation of VREfm sequencing data into routine outbreak detection and investigation.

## Methods

**Study design and data collection.** This project used data arising from the "Controlling Superbugs" study; a 15-month (April—June 2017[16] and October 2017—November 2018) prospective study that aimed to investigate the use of genomics to predict in-hospital multidrug-resistant organism transmission for a range of species, including *vanA* VREfm. Eight hospital sites across four hospital networks were involved in the study, resulting in 346 *vanA* VREfm isolates (308 patients) sent for WGS (pilot study[16] and implementation study[17]). Isolates were collected from patient samples (either clinical or screening samples) collected routinely from hospital inpatients (>24 h) at any time during their admission. Duplicate isolates were excluded as defined by; screening isolates of the same type (species and multilocus sequence type) already included in the study (first isolate collected included) or clinical isolates of the same type collected within 14 days of the previous clinical sample. An overview of the VRE screening and infection control practices can be found in Supplementary Table 1. Approximately two years of data detailing patient movements within and between wards were collected for all patients and the temporospatial overlaps for each patient pair determined. The sample collection strategy, sequencing and epidemiological data collection and categorisation are all as previously described[16]. Briefly, patient pairs were classified according to previously published definitions[18] as 'probable transmission' if on the same ward at the same time, 'possible transmission' if admitted to the same ward at a different time (within 60 days), or admitted to the same hospital at the same time; all other patients were classified as 'unlikely transmission'. An overview of the epidemiological data categorisation decision tree is provided in Supplementary Fig. 10.

**Whole-genome sequencing.** Genomic DNA was extracted from bacterial isolates using a JANUS automated workstation (PerkinElmer) and Chemagic magnetic

bead technology (PerkinElmer). Genomic DNA libraries were then prepared using the Nextera XT kit according to the manufacturer's instructions (Illumina Inc.). Whole-genome sequencing was performed on the Illumina NextSeq platform using 2 ×150 bp paired-end chemistry as before[16].

**Multilocus sequence typing (MLST).** All isolates underwent in silico MLST, to determine their sequence type (ST), using the *mlst* tool (v2.19.0) (https://github.com/tseemann/mlst) and the pubMLST database (https://pubmlst.org)[19] for *E. faecium*.

**Core genome MLST (cgMLST) and clustering.** cgMLST alleles of each isolates were defined using the public *E. faecium* cgMLST scheme, created by de Been et al[20]. and chewBBACA (v2.0.16)[21], implemented locally in the COREugate pipeline (v2.0.4) (https://github.com/kristyhoran/Coreugate). This pipeline determines the alleles of each core gene for every isolate as defined by the specific pathogen scheme. The *E. faecium* cgMLST scheme contains 1,423 genes. The number of allelic differences between each isolate within this core set of genes is then determined. The cgMLST clusters were then determined using single linkage clustering and a pairwise allelic difference threshold of ≤ 25. This threshold was determined based on the distribution of pairwise allelic differences and the inter-MLST pairwise allelic difference (see results).

**Reference genome selection.** The MLST core alignments (MLSTCA) used an "MLST reference" (published local complete genomes of the same ST). The cgMLST core alignments (cgMLSTCA) also used an "MLST reference" based on the ST of the cgMLST cluster. The cgMLST within-cluster reference core alignments (cgMLSTCRCA) used a "cgMLST cluster reference" (first isolate collected from each of the cgMLST clusters). A complete list of reference genomes is available in Supplementary Data 2. The method used to assemble all cgMLST cluster references is listed below.

**cgMLST cluster reference sequencing and assembly.** The earliest collected isolate from each of the cgMLST groups with more than two isolates were chosen for Oxford Nanopore sequencing (GridION). These isolates are listed in Supplementary Data 2. Genomic DNA was extracted using the Sigma-Aldrich GenElute Bacterial Genomic DNA kit according to the manufacturer's instructions. 1D Native Barcoding + Ligation Sequencing Kit (Oxford Nanopore) were used for library preparation and the GridION X5 (cell FLO-MIN106D R9) (Oxford Nanopore) was used for sequencing according to the manufacturer's instructions. Reads were filtered using porechop (v.0.2.4) (https://github.com/rrwick/Porechop) (barcode threshold 90 and two barcodes required) and filtlong (v.0.2.0) (https://github.com/rrwick/Filtlong) (minimum length 1000 bp and keep percent 95). Reads were then assembled using the Trycycler pipeline (v.0.3.3) (https://github.com/rrwick/Trycycler). This involved first subsampling the reads based on an estimated genome size of 3 Mb using the default method provided by Trycycler. Three different assemblers were used to generate nine draft assemblies (three with each assembler) – flye (v2.8.1)[22], miniasm+minipolish (v2.17) (https://github.com/lh3/miniasm)[23] and raven (v1.1.0) (https://github.com/lbcb-sci/raven). All assemblers were used with default settings. The contigs in all assemblies were then clustered and any cluster containing contigs from six or more assemblies proceeded to the reconcile step. Any contigs that prevented the reconcile step from completing were removed from the cluster. Following the reconcile, sequence alignment and the consensus step, the consensus sequences were polished with Illumina short reads using snippy (v4.6.0) (https://github.com/tseemann/snippy), the same program used in the core and pairwise alignments. Two rounds of polishing with short read data were completed with bcftools (v1.9) (https://github.com/samtools/bcftools) being used to edit the consensus sequence based on the SNPs identified using snippy.

**Phylogenetic trees.** When generating the species phylogenetic tree, three isolates were excluded from the study isolate core alignment analysis due to their divergence from the rest of the study isolate (Supplementary Fig. 11a). To better understand why these isolates were so divergent, the study isolates were put into context with global isolates. These global isolates randomly selected from a recent study by van Hal et al[24]. (n = 297) and had been divided into clade A1, A2 and B based on their genomic similarity. After constructing a phylogenetic tree using study and global context isolates, the divergent study isolates were found to cluster with clades A2 and B (Supplementary Fig. 11b). The study outliers and global context isolates are listed in Supplementary Data 1. The reference used for the species and global context trees was CP027497 (ST1421) [https://www.ncbi.nlm.nih.gov/genome/871?genome_assembly_id=368493] and it was chosen because it was the same ST as most study isolates and was also collected locally. Phylogenetic trees for each of the major STs is contained in Supplementary Fig. 12 and the references for the corresponding core alignments are in Supplementary Data 2. All phylogenetic trees were inferred using IQtree (v1.6.12)[25] with constant sites, 1000 bootstraps and a generalised time-reversible model of evolution (GTR + G4). For the species alignment, the core alignment length was 1,059,458 base pairs, the core SNP alignment was 4,020 sites and the core SNP alignment after masking for recombination using Gubbins[26] was 39 sites. All trees were made

using core SNP alignments that did not have recombination masking due to the highly variable effect that omitting recombination regions can have on pairwise distances[7] and the small size of the resulting core alignment. All trees were mid-point rooted and visualised in R (v4.0.2, https://www.R-project.org/) using phangorn (v2.5.5)[27], ape (v5.4)[28], ggtree (v2.3.4)[29] and ggplot (v.3.3.2)[30].

**Genomic comparisons.** Only isolates from the four major STs (ST1421, ST1424, ST80 and ST203) with more than three isolates in their respective cgMLST clusters were used in the genomic comparisons (n = 278). The four major STs were chosen as they comprise 88% of all study isolates. Only cgMLST clusters with three or more isolates can be used in subsequent analyses due to the requirements of the core genome alignments. An overview of the following methods is shown in Fig. 6. Isolate and reference details can be found in Supplementary Data 1 and 2.

**Core genome alignments.** Multi-locus sequence type core alignments (MLSTCA) were performed on all isolates in the same ST using a reference of the same ST. Core genome MLST core alignments (cgMLSTCA) were performed on all cgMLST clusters with greater than two isolates using reference genome of the same ST. Core genome MLST with cluster reference core alignments (cgMLSTCRCA) were core alignments performed on all cgMLST clusters with greater than two isolates using a reference of the same cgMLST cluster. All core genome alignments were performed using snippy (v4.6.0) (https://github.com/tseemann/snippy) with minfrac set to 0.9 and mincov set to 10. Pairwise SNP distances were calculated in R using harrietr (v0.2.3, https://github.com/andersgs/harrietr). Recombination masking was not performed for any core alignments to allow for comparison between all methods (PCDR and SKA cannot be screened for recombination). It has also been shown that recombination screening can have a highly variable effect on isolate pairwise distances, often inflating the number of closely related pairs[7]. An overview of these methods is shown in Fig. 6 and a list of all reference genomes can be found in Supplementary Data 2.

**PCDR (Pairwise comparison using *de novo* references).** Reads were assembled using SKESA[31] (v2.3.0) with default settings. SKESA was chosen due to its speed and high sequence quality. Mapping and variant calling was performed using snippy (v4.6.0) with minfrac set to 0.9 and mincov set to 10. Firstly, each isolate read set was first mapped to its corresponding SKESA assembly to identify any "self SNPs" and these "self SNPs" were removed from any subsequent pairwise comparisons based on their location in each contig. A breakdown of the number of "self SNPs" identified can be found in Supplementary Fig. 13. Next, SKESA assemblies were used as the *de novo* references for the pairwise comparison, with the reads from each isolate in the pair, being mapped to the *de novo* assembly of the other (non-self) isolate. The average number of SNP variants (excluding any "self SNPs"), calculated from these two alignments, was used as the pairwise SNP distance between two isolates. All PCDR distances presented and referenced in the paper are those that were adjusted for "self SNPs". PCDR was used as the gold standard for determining isolate relatedness, as this method is consistent with the traditional mapping and core alignment approach but also maximises the proportion of genome considered in each pairwise alignment (Fig. 4).

**SKA (split kmer analysis).** The SKA (v1.0) package[32] was used to generate kmer files (k = 15) for each of the input isolates from the short-read data, using the fastq subcommand (default thresholds were used for all variables). Isolate-to-isolate pairwise SNP distances were then calculated using ska distance with default settings. When using ska distance, a SNP is defined as "Number of split kmers found in both samples where the middle base is an A, C, G or T but differs between files". In contrast to a traditional core genome alignment, SKA compares isolate pairs based on all genomic data contained in the read files, which is equivalent to a whole-genome comparison.

**SNP distance threshold for transmission inference.** To determine an appropriate SNP distance threshold for putative transmission links, we assessed the level of genetic diversity between isolate pairs sampled from the same patient for each of the genomic methods (intrapatient diversity). Only isolate pairs arising from the same patient and from within the same cgMLST cluster were considered (n = 30 isolate pairs) to avoid comparing distant isolates from different acquisition events. Within patient pairs from different cgMLST clusters (n = 10) were shown to have a much greater SNP distance using SKA (distances ranging from 396 to 3176 SNPs) than those of the same cgMLST cluster (0–34 SNPs). These pairwise distances can be found in Supplementary Data 3. A threshold of ≤90 days between isolate pair dates of collection was subsequently used to exclude temporally distant outliers (n = 2 isolate pairs excluded) (Supplementary Fig. 5a). The pairwise SNP distances for the remaining 28 isolate pairs were calculated for all four genomic methods assessed (cgMLSTCA, cgMLSTCRCA, PCDR and SKA). The pairwise SNP distance representing the 90th percentile for each of these methods (Supplementary Fig. 5b) was used as the threshold for putative transmission links for inter-patient isolate pairs. An overview of this process is shown in Supplementary Fig. 14.

**Hospital case study**. Vancomycin-resistant *Enterococcus faecium* isolates collected between January 2021 and July 2021 from a single hospital as part of standard practice were sequenced as above. These isolates were a combination of screening and clinical samples. This data set included 50 *Enterococcus faecium* isolates (a combination of VanA, VanB and vancomycin susceptible enterococci) from 45 patients. Isolates were assessed for genomic transmission links using cgMLST clustering (pairwise allelic difference of ≤25) and SKA. A pairwise genomic distance of ≤7 SNPs was used to define SKA genomic transmission links. Any pairs that were identified as being genomically linked were then assessed for epidemiological links. Bed move data was collected for all patients between July 2019 and July 2021 (2 years). The same decision tree that was used in the main study was used to categorise the degree of epidemiological links, except as all isolates in the case study were from the same hospital, the hospital unit was used instead of hospital to provide greater resolution in links (Supplementary Fig. 10).

**Data visualisation and statistics**. Figures were generated in R (v4.0.2, https://www.R-project.org/) using tidyverse[33], networkD3[34], naniar[35], ggpubr[36], patchwork[37], htmlwidgets[38] and htmltools[39]. Flow diagrams were generated using LucidChart (https://www.lucidchart.com). The coefficient of determination ($R^2$) was calculated using ggpmisc[40] and was based on a linear regression model.

**Ethics approval**. The "Controlling Superbugs" study was approved by the Melbourne Health Human Research Ethics Committee (HREC) and endorsed by the corresponding HREC at each participating site (HREC/13/MH/326). This ethical approval also covered the hospital case study.

**Reporting Summary**. Further information on research design is available in the Nature Research Reporting Summary linked to this article.

## Data availability

Illumina sequencing reads for all samples from the formal "Controlling Superbugs" were deposited into GenBank under BioProject PRJNA565795. Illumina sequence reads for all samples part of the case study were deposited into BioProject PRJEB49226. Reference assemblies used can be found in BioProject PRJNA565795 or PRJNA433676. A full isolate list and associated metadata can be found in Supplementary Data 1. Supplementary Data 2 contains a list of all reference genomes used in the study and summary statistics for the core genome alignments. Genetic distance data for the method comparison and ward move data can be found in Supplementary Data 3. An isolate list, associated metadata and ward move data for the hospital case study can be found in Supplementary Data 4. Only the processed ward move data is available in the supplementary data, the raw ward move data are not available due to data privacy laws. The deidentified raw ward move data are available upon request from the corresponding author (BPH) and will be actioned within 1 month. There are no conditions of access and the data can be used freely for research activities.

## Code availability

All code used for the genomic comparisons can be found in Supplementary Data 5.

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

## Acknowledgements

The authors would like to acknowledge the other members of the Controlling Superbugs Study Group include Robyn Lee (previously MDU, currently University of Toronto), Rhonda Stuart (Infectious Diseases, Monash Health; Medicine, Monash University), Tony Korman (Infectious Diseases and Microbiology, Monash Health; Medicine, Monash University), Caroline Marshall (VIDS, Melbourne Health; Peter Doherty Institute), Hiu Tat (Mark) Chan (Microbiology, Melbourne Health), Maryza Graham (Infectious Diseases and Microbiology, Monash Health; Medicine, Monash University), Marcel Leroi (Microbiology, Austin Health), Caroline Reed (Microbiology, Melbourne Health and Peter MacCallum Cancer Centre), Michael Richards (VIDS, Melbourne Health; Peter Doherty Institute), Monica Slavin and Leon Worth (Infectious Diseases, Peter MacCallum Cancer Centre; National Centre for Infections in Cancer; University of Melbourne), Elizabeth Grabsch (Microbiology, Austin Health), Joanna Price and Carolyn Tullett (Infection Control, Austin Health), Despina Kotsanas (Microbiology, Monash Health), Louise Wright (Infection Control, Monash Health), Suraya Hanim Abdullah Hashim and Jennifer Mitchell (Infectious Diseases, Melbourne Health), Olivia Smibert (Infectious Diseases, Peter MacCallum Cancer Centre), and Carol Wedge (data entry, Austin Health). This work was supported by the Melbourne Genomics Health Alliance (funded by the State Government of Victoria [Australia], Department of Health and Human Services, and the ten member organisations); a National Health and Medical Research Council (Australia) Partnership grant (GNT1149991) and individual grants from National Health and Medical Research Council (Australia) to NLS (GNT1093468), JCK (GNT1142613) and BPH (GNT1196103).

## Author contributions

B.P.H. and M.L.G. designed and managed the Controlling Superbugs Study. B.P.H., C.L.G., T.P.S., and C.H. designed this project and verified the underlying data of the study. C.H. conducted all genomic, bioinformatic and statistical analyses, and produced the manuscript and all accompanying figures and tables. N.L.S. was part of the Controlling Superbugs Study Group for the initial project, helped with data collection and quality control, provided guidance and input throughout, and edited the manuscript. J.C.K. was part of the Controlling Superbugs Study Group for the initial project. H.W. performed the long-read sequencing. T.S. and K.H. wrote various bioinformatic tools that were used in the study and provided bioinformatic advice. T.P.S. provided guidance and feedback both during the study and for the final manuscript. P.K., K.B., C.M., and D.A.W. provided data for the hospital case study, guidance on the results and edited the manuscript.

## Competing interests

The authors declare no competing interests.
