## [Peer Review File · Nature Communications]

Optimising genomic approaches for identifying vancomycin-resistant *Enterococcus faecium* transmission in healthcare settingsREVIEWER COMMENTS

Reviewer #1 (Remarks to the Author):

The paper by Higgs et al demonstrates that genomic epidemiology is evolving into a scientific expertise with its proper analytical methods and efforts toward standardization. This is much needed and as such the manuscript is worthy of publication. Technical normalization and the use of adequate reference genomes are important parameters for correct use of the NGS methodology. Still there are a couple of generic questions that come to mind. The authors largely ignore the how and why of the practical application of their methods. I realize this is not at the heart of their work but I think it should be made clear that all that is currently done in the Infection Control domain is solely retrospective in nature (strains used in the Higgs study date back more than three years already for instance). The only way that this technology is going to really contribute to prevention of outbreaks is if the technology can be used prospectively where WGS and data interpretation should be done overnight and result in a report that would be useful to infection control measures (and understandable by infection control practitioners). It would be nice if the authors could reflect on this a little, same for the differences in use of this methodology in open departments versus more closed units such as intensive care settings. A reflection on the use of this technology in emerging economies would be appreciated as well. Again, this is not really the main target of the paper but it is key to the ultimate implementation of the novel infection control workflow suggested by the authors. The other main issue I note with the paper is the lack of assessment of real life impact. The authors touch upon the subject between lines 477 and onward but to me the big question is whether the authors are using a Ferrari to plough a field ... Is there a risk of over-interpretation of the data when digging as deep as the authors do? If you set a cut-off of 7 SNPs then how do you interpret the presence of 8 or 9 SNPs? Is this really a highly analytical science or does one need a somewhat more relaxed, biological data interpretation? What is the clinical added value of applying the algorithm developed and how big is the chance of preventing transmission or even outbreaks in real life? To me it is not always completely clear throughout the manuscript which genomes are used as references for the various methods applied by the authors. Are the references always the same or different? Also, would there have been value in sequencing multiple colonies from individual cultures to assess for the intra-patient genomic diversity level? One final major remark, Pontinen et al recently published a similar paper for *E. faecalis* (Nature Communications 2021;12:1523) and it would be good to discuss the outcomes for this species as well. Below follow some more specific comments.

Lines 28-33: I think I understand what you are saying here but I am not completely sure. Especially the end of the sentence is confusing (...events than the kmer??).

Line 62: Data sounds plural to me so data are.

Line 114: I do not understand why duplicate screening isolates were eliminated. Assessing diversity between such isolates could help set a reference value for intrinsic variation in the genome of a strain causing persistent colonization or infection. Would be interesting to consider that in the context of patient treatment as well.

Line 116: ... data were ...

Lines 195-198: I understand the selection used by the authors from a genomic perspective. Eliminating the unique strains which are likely to be imported singletons to me sounds like throwing away a large chunk of the strains that in the end will be the most challenging from the infection control perspective. Where do these come from? Do they pose an outbreak risk? Are the major clones major clones since they are more outbreak prone than such singletons???

Line 226: Split kmer analysis with the m included I guess.

Line 236: Same comment as for line 114 above more or less? Why exclude these strains?

Line 246-251: I find this a scary hyper-jargon section since for a simple person such as myself this raises issues on the methods used: are these neutral or do they one way or another polish or massage data into something that looks very nice but may not be completely without bias No need to explain this in more detail but it tells me that clinical microbiologists and infection control practitioners may need additional bio-info education.

Line 280: As alluded to in the generic section above, would this be a cut-off that would be equal for all clinical settings (open-closed, developed versus developing economies, education levels, laboratory equipment, sample transport, sample heterogeneity, antibiotic treatment modules for

patients etc).

Lines 369-375: So in the end, what are the more decisive markers, the genomic ones or the more classical patient-related ones????

Line 382-384: Is what is considered better by population microbiologists also always better for infection control practitioners?

Line 393-395: In outbreak management strains that move from one to another patient may need a vector (personnel, visitors, hospital environment). Have you been able to consider this in the epi-study you performed? Were personnel or the environment ever sampled?

Line 604: "undefined"??

Figure 1a: Why are there quite some white regions in the outer circle?? Are these the singletons that are missing??

Figure 5: When one starts thinking the moment there would be a suspected outbreak then one would already be late in many cases. Maybe include a "prospective scenario" here, assuming that sequencing and genome typing can be done in like an hour??

Reviewer #2 (Remarks to the Author):

The manuscript by Higgs et al. uses a cohort of 308 prospectively collected vancomycin-resistant *E. faecium* isolates from the "Controlling Superbugs Study" to compare genomic approaches for the quantification of genetic relatedness of isolates in this cohort. Core genome alignment-based methods with increasing relatedness of reference genomes as well as pairwise comparison methodologies were evaluated based on the resolution and granularity of the population structure. The authors compared these methods to MLST and evaluated the accuracy of the identification of putative transmission clusters compared to the "gold standard" method with substantiation from some epidemiological data. Based on the results of these analyses, the authors recommended a "real world" workflow for identification of putative transmission events that uses core genome MLST (cgMLST) to broadly cluster with genetically similar core genomes, then perform split kmer analysis (SKA) on isolates within the cgMLST clusters. The conclusions are reasonably supported with both genomic and some epidemiological evidence. However, I have some major concerns:

1. While it is commendable that the authors are trying to set standards for this field, the methods used here have been previously extensively described and are applied to an extremely limited set of isolates, thus limiting the novelty and impact of the results obtained.
2. The analysis uses unusual epidemiologic definitions of transmissions that do not match the definitions leveraged in clinical assessments of outbreaks, making the conclusions difficult to sustain. A major issue is that the work, as presented, lacks a detailed validation of the SNP threshold determinations and clustering methods paired with robust epidemiological information. Since the goal is to help using these approaches for hospital epidemiology and infection control, the lack of validation precludes an interpretation of the results, diluting the impact of the work.
3. The numbering of the supplementary figures is very confusing. The main and supplementary figures should be re-numbered according to reference in the manuscript.
4. Consider adding the number of core genes used in the cgMLST scheme ($n=1,423$), as it further highlights the increased resolution cgMLST provides relative to MLST and helps put the pairwise allelic difference thresholds used in these analyses into perspective.
5. Lines 178-179: Could the authors comment on why snippy was chosen to polish consensus sequences? The documentation for this tool suggests that it is not ideal for this particular application.
6. Lines 185-187: There are major issues with figures. The authors need to provide more explanation regarding the exclusion of the three "outlier" isolates, particularly since one of the outliers is of the same ST as the reference used for the species tree (ST1421). Additionally, the authors should provide the rationale for the choice of reference for Figure 2 and indicate whether they masked for recombination.

7. Line 190: This tree (Fig 2) does not appear to be midpoint-rooted, as there should not be such extreme phylogenetic “outliers” (ST18) after rooting. Additionally, the fact that ST18, which is globally a relatively highly prevalent ST, appears to be quite genetically distinct from the rest of the cohort raises concerns about the generalizability of this cohort.

8. Lines 213, 226: Though implied and alluded to in Figure 2, it needs to be delineated in the text that these methodologies (PCDR, SKA) utilize the entirety of the genome, not just the core genome. This is an important distinction that may go unrecognized by a reader less familiar with these methods, affecting the interpretation of the results.

9. For Figure 1b –the two panels should be put on the same axes so they are directly comparable. The subpanel should not be restricted to 50 differences, as it may confuse a reader and make them think the largest distance is 50.

10. I have substantial concerns about making claims about the utility of these methods in identifying transmission when there are not any specific real-world applications that follow up on the detected potential/likely transmission events. Indeed, confirmation of spatiotemporal epidemiologic links that strengthens the assumptions is needed to draw meaningful conclusions. The whole paper hinges on this method being applicable to real-time investigations, but there is not evident application demonstrated in the manuscript.

11. I am not sure that the proposed workflow can function as a “one size fits all” approach. For moderately sized cohorts such as the one analyzed in the paper, this approach is reasonable, but for relatively small cohorts, such as in localized hospital outbreak settings, might it be more appropriate to use SKA as an initial clustering step instead of cgMLST, then use PCDR for the fine-resolution final step of only the most highly-related isolates. This approach would limit the computational resources necessary, and it would also enable full genome resolution of the most related isolates taking the genetic context into consideration.

Reviewer #3 (Remarks to the Author):

In this paper, Higgs et al. assess different genomic approaches to identify vancomycin-resistant *Enterococcus faecium* transmission. Find below a summary of my major comments followed by all comments.

Summary of major comments:

- The authors state that “thresholds [in the literature] are not accompanied by methods to standardise the preceding steps in the analyses”. The authors should make the commands they used available to facilitate adoption of their methods.
- It is key for interpretation of the results that the authors include further information on the study design in the Methods section. Also, they should be more explicit on how epidemiological data is used to define probable and possible transmission events. See related comments below.
- The authors specify that PCDR (Pairwise comparison using de novo references) is “the gold standard for determining isolate relatedness”. However this method could be compromised by inflated SNP counts due to error in the de novo assembled sequence used as reference. The authors must ensure and demonstrate that PCDR distances are not inflated by miss-assemblies.
- The authors should assess the impact of recombination on genetic distances, particularly given the lack of correlation between the SNP distances of different methods.

- In Supplementary Figure 11, the intra-patient diversity is rather comparable between methods (6 to 13 SNPs), but much higher for cgMLSTCA, which results in more than twice as many putative transmission links compared to other methods. How can this be explained especially when compared to cgMLSTCRCA?

Comments

Abstract - line 24 "methods are not yet optimised". It is not clear what the methods need to be optimised for.

Methods

On the study design and data collection. The authors should include more information on the screening procedures in place at the eight hospitals. 308 positive patients seems a rather low number considering the duration of the study (15-months) and the number of participating hospitals (eight hospitals). However, this is hard to assess without knowing the exact VRE screening procedures, what wards and patient populations were targeted and the VREfm positivity obtained. At the moment it looks as if a very sparse sampling took place, which would limit the detection of transmission.

Lines 116 to 120. In statement: "in putative transmission as defined by the "Controlling Superbugs" study (patients with an isolate determined to be genomically related to another isolate given the set threshold), and the temporospatial overlaps for each patient pair determined." It is crucial to define how putative transmission were defined from epidemiological data in this manuscript.

Lines 121 to 123: "sample collection strategy, sequencing and epidemiological data collection and categorisation" need to be brought into this manuscript, as they are key for interpretation.

Line 155. Can the authors check if Supplementary Data 1 is the one containing the "complete list of reference genomes" as specified here? Supplementary Data 1 seems to be the table containing isolate details and Supplementary Data 2 the one with the list of reference genomes.

Line 161. In statement "These isolates are listed in supplementary data." Please indicate what specific supplementary data file.

Lines 177 - 179. Did the authors map Illumina short reads to the consensus sequences using Snippy? If so, they need to be more explicit about the use of Illumina short reads in this statement for clarity.

Line 195. In statement: "Only isolates from the four major STs (ST1421, ST1424, ST80 and ST203) with more than three isolates in their respective cgMLST clusters were used in the genomic comparisons (n=278)." It is not clear to this reviewer the rationale for excluding non-major STs from genomic comparisons.

It would be informative to specify what proportion of the genome is being used for comparisons by each method. SRA and PCDR use 100% of the genome in pairwise comparisons; but what percentage do cgMLSTCA and cgMLSTCRCA use?

PCDR (Pairwise comparison using de novo references). Lines 214 to 224. Mapping the short reads of one isolate against the de novo assembly of another could produce inflated SNPs counts due to miss-assemblies in the de novo assembly used as reference. Have the authors attempted to map the short reads of each isolate against its own SKESA assembly? In other words, what steps have the authors undertaken to make sure mis-assemblies are not producing spurious SNPs? This is particularly important as authors are using PCDR as "the gold standard for determining isolate relatedness".

SKA (split kmer analysis). Lines 226 to 230. More information on how the ska distance is calculated is needed here. Is it the number of k-mer mismatches?

SNP distance threshold for transmission inference. Lines 232 - 244. Determining an appropriate SNP distance threshold for transmission analyses based on intra-patient diversity is an established approach. But first, the clonality of isolates from the same patient must be confirmed (to avoid comparing distant isolates from different acquisition events). Did the authors check that isolates from different cgMLST in the same patient were much more distant (in terms of number of SNPs) than isolates from the same cgMLST?

In Supplementary Figure 11, the intra-patient diversity is rather comparable between methods (6 to 13 SNPs), but much higher for cgMLSTCA; how can this be explained?

Results

Figure 1. It would be advisable to root the phylogenetic tree. The authors should assign *E. faecium* isolates to clade A or B. Most likely, the largest clade in Supplementary Figure 10 corresponds to clade A, as this is the most commonly isolated clade among hospital isolates, but this needs to be confirmed with labelled contextual isolates. The two outliers in red at the top of this figure might correspond to clade B or basal clade A isolates. If that's the case, these isolates can be used as an outgroup to root the clade A.

In this section, the authors compare the agreement between MLST and cgMLST clusters with the population structure, as defined by the phylogenetic tree. Recombination needs to be detected and removed prior to building the phylogenetic tree (using tools such as Gubbins), to obtain a robust phylogenetic tree.

MLST is known not be a robust genotypic scheme for *E. faecium*, the authors should cite studies reporting this. Indeed, it is expected for some major STs to be polyphyletic, i.e. they do not fall within single monophyletic clades. Most cgMLST clusters seem to be monophyletic, which is good, with the exception of cgMLST cluster 6 and 17. It would be good to check if they become monophyletic once a recombination-free and rooted phylogenetic tree is used.

The use of a 25 cgMLST SNP threshold to define clusters seems reasonable, given the distribution of cgMLST SNP distances shown in Figure 1b.

Line 318 - 319. In statement "pairwise SNP distance ≤ 50 ". Please indicate what type of SNP distances, of the different ones calculated in this work, are referred to here.

Section "For very closely related isolate pairs, direct pairwise comparison tools should be used". Lines 302 - 332. I found unexpected that core-genome SNP distances do not correlate with whole-genome distances (i.e. PCDR distances). As explained in a point above, the authors must ensure and demonstrate that PCDR distances are not inflated by miss-assemblies. It would be interesting to explore if the lack of correlation is due to recombination, which again, is expected to inflate SNP counts. As recombination is expectedly lower, or absent, among highly related strains that diverged recently, it would be interesting to test for correlation separately among highly and distantly related pairs.

Lines 337 to 344. As pointed above, in Supplementary Figure 11, the intra-patient diversity is rather comparable between methods (6 to 13 SNPs), but much higher for cgMLSTCA, which results in more than twice as many putative transmission links compared to other methods. How can this be explained especially when compared to cgMLSTCRCA?

Lines 367 to 375.

372 to 375. The definitions of epidemiological evidence (i.e. probable and possible transmission likelihood) need to be included in the methods, as it is key for interpretation of these results.

Figure 4 and Supplementary Figure 9. In addition to reporting the proportion of isolate pairs that are genomically linked under the different methods/thresholds, it would be relevant and more intuitive to report the proportion of patients that are genomically linked too.

Please make sure the numbering of supplementary figures is consistent with the order they appear in the manuscript.

REVIEWER COMMENTS

Reviewer #1 (Remarks to the Author):

The paper by Higgs et al demonstrates that genomic epidemiology is evolving into a scientific expertise with its proper analytical methods and efforts toward standardization. This is much needed and as such the manuscript is worthy of publication. Technical normalization and the use of adequate reference genomes are important parameters for correct use of the NGS methodology. Still there are a couple of generic questions that come to mind. The authors largely ignore the how and why of the practical application of their methods. I realize this is not at the heart of their work but I think it should be made clear that all that is currently done in the Infection Control domain is solely retrospective in nature (strains used in the Higgs study date back more than three years already for instance). The only way that this technology is going to really contribute to prevention of outbreaks is if the technology can be used prospectively where WGS and data interpretation should be done overnight and result in a report that would be useful to infection control measures (and understandable by infection control practitioners). It would be nice if the authors could reflect on this a little, same for the differences in use of this methodology in open departments versus more closed units such as intensive care settings.

Response: *These are all great points and the ability to use the proposed pipeline for prospective outbreak detection was very high on our priority list. We first had to ensure that the pipeline would produce accurate and reliable data and to do this we used retrospective data. However, we have now updated our manuscript to also include a hospital case study from the first half of 2021. Use of our proposed analysis method in this case study produced useful and actionable data at the hospital level, demonstrating that implementation of the approach into routine practice is achievable (Lines 308-322, 464-484, 603-619).*

A reflection on the use of this technology in emerging economies would be appreciated as well. Again, this is not really the main target of the paper but it is key to the ultimate implementation of the novel infection control workflow suggested by the authors.

Response: *This is an important point, and has been added to the discussion, with a mention about how well-defined thresholds and standardised pipelines makes these kinds of methods more easily adaptable to settings with less resources or training (Line 593-596).*

The other main issue I note with the paper is the lack of assessment of real life impact. The authors touch upon the subject between lines 477 and onward but to me the big question is whether the authors are using a Ferrari to plough a field ... Is there a risk of over-interpretation of the data when digging as deep as the authors do? If you set a cut-off of 7 SNPs then how do you interpret the presence of 8 or 9 SNPs? Is this really a highly analytical science or does one need a somewhat more relaxed, biological data interpretation? What is the clinical added value of applying the algorithm developed and how big is the chance of preventing transmission or even outbreaks in real life?

Response: *This is a very valid point and can be a danger with transmission analysis methods. What we aimed to show here is that the conventional methods often result in the opposite problem of labelling everything as being involved in transmission and not allowing any meaningful interpretations about wards or units that are involved. Instead, when using our proposed method there is a much greater level of resolution allowing identification of specific wards that could be the location of the transmission event. The point is well made about the interpretation of SNP distances close to the cut off, but we opted for a model that was more standardised and required little interpretation, allowing it to be easily automated as well as making it more understandable for infection control practitioners. We also performed an additional analysis on data from a recent study by Gouliouris et al. (Nature Microbiology 2021;6:103-111) that sequenced multiple colonies from the same sample and*

observed an inpatient diversity level of no more than 5 SNPs in 95% of comparisons when using SKA. A threshold very similar to our own of 7 SNPs, adding validity to the threshold. This is explored in the discussion in lines 559-578.

To me it is not always completely clear throughout the manuscript which genomes are used as references for the various methods applied by the authors. Are the references always the same or different?

Response: *The references change depending on the method used. For each of the core genome alignment methods that use all isolates in a cgMLST cluster, they are either of the same MLST (cgMLSTCA) or same cgMLST cluster (cgMLSTCRCA). The reference for each of the groups can be found in Supplementary Data 2. For PCDR and SKA pairwise comparisons are used and so every isolate is effectively a reference. We do understand that the number of methods used in the paper makes it complex, so we have included method flow diagrams and schematics to make it as clear as possible (Figures 2 and 5). In the discussion we have also provided brief summaries of each method when referring to them to further aid understanding (lines 4515-557).*

Also, would there have been value in sequencing multiple colonies from individual cultures to assess for the intra-patient genomic diversity level?

Response: *This is a good suggestion and would have increased the accuracy with which we could assess the genomic diversity within each patient. The thresholds that we defined were like those that have been previously described as referenced in the discussion (Line 537-540). In addition, we used *E. faecium* sequence data from a recent study by Gouliouris et al. (Nature Microbiology 2021;6:103-111) that did sequence multiple colonies from the same sample and observed an inpatient diversity level of no more than 5 SNPs in 95% of comparisons when using SKA (Line 571-578).*

One final more major remark, Pontinen et al recently published a similar paper for *E. faecalis* (Nature Communications 2021;12:1523) and it would be good to discuss the outcomes for this species as well. Below follow some more specific comments.

Response: *Thank you for this suggestion. The paper referenced above is more focused on population and evolutionary genomics with analyses at a species level over a long time period. Given our paper is specifically focussing on hospital transmission, requiring very high-resolution genomics over short time periods, the findings from this paper were not found to be directly relevant. We therefore have decided not to include it.*

Lines 28-33: I think I understand what you are saying here but I am not completely sure. Especially the end of the sentence is confusing (...events than the kmer??).

Response: *This section has been updated and now reads “We showed that pairwise genome comparisons using a kmer-based approach provided the high level resolution needed to infer patient-to-patient transmission, whereas the more common mapping-to-reference-genome was not sufficiently discriminatory, defining more than three times more genomic transmission events than the kmer-based approach (3729 compared to 1079 events).” (Lines 36-41).*

Line 62: Data sounds plural to me so data are.

Response: *Changed main text to “data are”.*

Line 114: I do not understand why duplicate screening isolates were eliminated. Assessing diversity between such isolates could help set a reference value for intrinsic variation in the

genome of a strain causing persistent colonization or infection. Would be interesting to consider that in the context of patient treatment as well.

Response: *This is a good point, however the exclusion of these duplicate isolates was part of the original protocol for the Controlling Superbugs study and therefore they were not available for this paper. The text has been updated to include more detail on what a “duplicate isolate” was (Line 128-132).*

Line 116: ... data were ...

Response: *Update in main text*

Lines 195-198: I understand the selection used by the authors from a genomic perspective. Eliminating the unique strains which are likely to be imported singletons to me sounds like throwing away a large chunk of the strains that in the end will be the most challenging from the infection control perspective. Where do these come from? Do they pose an outbreak risk? Are the major clones major clones since they are more outbreak prone than such singletons???

Response: *The main text has been updated to make the selection more understandable (Lines 234-240). Although singletons were not used in the genomic comparison workflow it does not exclude them from being included when the workflow is implemented. Major STs were focussed on as these represent most infections and are more prone to misinterpretation due to their varying levels of diversity and the fact that larger sample sizes are more likely to influence core genome size in core genome alignments.*

Line 226: Split kmer analysis with the m included I guess.

Response: *Updated in main text.*

Line 236: Same comment as for line 114 above more or less? Why exclude these strains?

Response: *The aim was to assess the level of diversity within a patient and use that as a threshold to assess transmission. We chose not to use any same patient pairs that were from different STs as this implies they are likely co-colonised with different strains. If we had kept all isolates from the same patients, regardless of ST, this would have drastically increased the transmission inference threshold and would have also been difficult to calculate due to many genomic diversity methods relying on MLST groups. The isolates collected more than 90-days apart were excluded to reduce any effect of temporality on SNPs, rather than diversity at a singular time point. The time between isolates is shown in Sup Figure 4 a) and the 90-day threshold meant two pairs were excluded.*

Line 246-251: I find this a scary hyper-jargon section since for a simple person such as myself this raises issues on the methods used: are these neutral or do they one way or another polish or massage data into something that looks very nice but may not be completely without bias No need to explain this in more detail but it tells me that clinical microbiologists and infection control practitioners may need additional bio-info education.

Response: *We feel it is appropriate to include all information regarding packages used to make the visualisations and perform the statistical analysis, however it is not necessary to understand how all these packages and processes work to use the pipeline described; many are not required for typical use of the pipeline.*

Line 280: As alluded to in the generic section above, would this be a cut-off that would be equal for all clinical settings (open-closed, developed versus developing economies,

education levels, laboratory equipment, sample transport, sample heterogeneity, antibiotic treatment modules for patients etc).

Response: *We agree with this point, and yes, this would be a cut off that is applicable across all clinical settings. The aim of this paper is to present a pipeline for standardising transmission analysis for VRE in the hospital environment, so it is important that methods and cut offs be the same across all settings to allow for comparison.*

Lines 369-375: So in the end, what are the more decisive markers, the genomic ones or the more classical patient-related ones????

Response: *These markers go hand in hand, and both are needed to get a true understanding of transmission. However, due to the complexities surrounding the extraction of epidemiological data and the time it takes to prepare, use of genomic analysis to provide clear targets for further epidemiological investigation allows for a much more effective use of resources. Through our work at the hospital interface it has become clear that this approach will provide an ideal use of resources in identifying transmission in these settings. This is now discussed in relation to the hospital case study in lines 603-619.*

Line 382-384: Is what is considered better by population microbiologists also always better for infection control practitioners?

Response: *Yes, if infection control practitioners are using said data to make inferences about transmission and isolate relatedness. Clustering and grouping come with their pitfalls but here we aim to present a pipeline that will satisfy the population microbiologist while also allowing interpretation and action by infection control practitioners.*

Line 393-395: In outbreak management strains that move from one to another patient may need a vector (personnel, visitors, hospital environment). Have you been able to consider this in the epi-study you performed? Were personnel or the environment ever sampled?

Response: *Vectors and personal were not sampled during the duration of the controlling superbugs study, therefore, we cannot make inferences about how important they were. However, if these vector samples had been taken, they could be included in the pipeline just as the patient isolates have been and interpretations about likely transmission events could also be made.*

Line 604: “undefined”??

Response: *Fixed in reference list*

Figure 1a: Why are there quite some white regions in the outer circle?? Are these the singletons that are missing??

Response: *Singletons are shown in white, and this is clarified in the legend of the figure 1.*

Figure 5: When one starts thinking the moment there would be a suspected outbreak then one would already be late in many cases. Maybe include a “prospective scenario” here, assuming that sequencing and genome typing can be done in like an hour??

Response: *This is a great point and figure 5 has been updated to describe the scenario as “Active Outbreak Detection” to reflect the prospective scenario in which this pipeline can be used.*

Reviewer #2 (Remarks to the Author):

The manuscript by Higgs et al. uses a cohort of 308 prospectively collected vancomycin-resistant *E. faecium* isolates from the “Controlling Superbugs Study” to compare genomic approaches for the quantification of genetic relatedness of isolates in this cohort. Core genome alignment-based methods with increasing relatedness of reference genomes as well as pairwise comparison methodologies were evaluated based on the resolution and granularity of the population structure. The authors compared these methods to MLST and evaluated the accuracy of the identification of putative transmission clusters compared to the “gold standard” method with substantiation from some epidemiological data. Based on the results of these analyses, the authors recommended a “real world” workflow for identification of putative transmission events that uses core genome MLST (cgMLST) to broadly cluster with genetically similar core genomes, then perform split kmer analysis (SKA) on isolates within the cgMLST clusters. The conclusions are reasonably supported with both genomic and some epidemiological evidence. However, I have some major concerns:

1. While it is commendable that the authors are trying to set standards for this field, the methods used here have been previously extensively described and are applied to an extremely limited set of isolates, thus limiting the novelty and impact of the results obtained.

Response: *Although the methods themselves have been described elsewhere they have not been directly compared in this way with such detailed epidemiological data, providing clear novelty to this paper. The set of isolates may be somewhat limited from a worldwide VRE population view but from a hospital outbreak point of view, these isolates are extensive comprising 15 months of isolates across 4 hospital networks. In addition to using retrospective data to develop a comprehensive analysis pipeline, we have now also trialled implementing the pipeline into practice to provide more recent data with outcomes directly affected by the results we obtained (see details of hospital case study; lines 464 – 484 and 603-619).*

2. The analysis uses unusual epidemiologic definitions of transmissions that do not match the definitions leveraged in clinical assessments of outbreaks, making the conclusions difficult to sustain. A major issue is that the work, as presented, lacks a detailed validation of the SNP threshold determinations and clustering methods paired with robust epidemiological information. Since the goal is to help using these approaches for hospital epidemiology and infection control, the lack of validation precludes an interpretation of the results, diluting the impact of the work.

Response: *The epidemiological definitions used have been previously published (Holt et al. PLOS ONE 2016 11:7) and were the same as those used in the initial Controlling Superbugs study (Sherry et al. Infection Control Hospital Epidemiology 2021 42(5):573-81). The clustering methods as well as the SNP thresholds are similar to those previously published as referenced in the discussion (line 559-562). To alleviate some concern surrounding the effect of sequencing multiple colonies as raised by another reviewer, we have also used data from a recent study by Gouliouris et al. (Nature Microbiology 2021;6:103-111) that did sequence multiple colonies from the same sample and observed an inpatient diversity level of no more than 5 SNPs in 95% of comparisons when using SKA. A threshold very similar to our own of 7 SNPs (Line 573-578). In addition, we have now also included a hospital case study that uses the proposed method to identify transmission clusters (identification of clusters was done based on genomics only). These transmission clusters were supported by epidemiological evidence (Figure 6 and Sup Figure 14).*

3. The numbering of the supplementary figures is very confusing. The main and supplementary figures should be re-numbered according to reference in the manuscript.

Response: *The numbering of the supplementary figures has been updated to reflect how they appear in the manuscript.*

4. Consider adding the number of core genes used in the cgMLST scheme (n=1,423), as it further highlights the increased resolution cgMLST provides relative to MLST and helps put the pairwise allelic difference thresholds used in these analyses into perspective.

Response: *Thank you for this suggestion. The number of genes in the cgMLST scheme has been added to methods, results and Figure 1 legend (Line 163, 356 and 829).*

5. Lines 178-179: Could the authors comment on why snippy was chosen to polish consensus sequences? The documentation for this tool suggests that it is not ideal for this particular application.

Response: *Snippy was used for polishing as it is the tool that was also used for identifying SNPs in the core alignment analysis. By using it for the polishing step we can be sure that there will be no self SNPs identified between the completed reference genome and the short-read data. A comment on this has also been added to the relevant methods section (Line 202).*

6. Lines 185-187: There are major issues with figures. The authors need to provide more explanation regarding the exclusion of the three “outlier” isolates, particularly since one of the outliers is of the same ST as the reference used for the species tree (ST1421). Additionally, the authors should provide the rationale for the choice of reference for Figure 2 and indicate whether they masked for recombination.

Response: *These isolates were excluded due to their comparative distances from the rest of the study isolates as well as their position in the global context of VRE (this is now shown in an updated sup. Figure 2b). The excluded isolates were found to be associated with clade A2 and B, while the remaining isolates were associated with clade A1. The reviewer raises a good point that the excluded isolate is also the same as the reference and this adds to points made in the paper about the diversity within the STs. The tree was made using an alignment that was not masked for recombination and this is now explicitly stated in the figure 2 legend. We have also now included more information about the rationale for not masking for recombination and the reason for excluding the divergent isolates in lines 208 to 231. The reference was chosen because it was the same ST as most isolates and was also collected locally (Line 170-177).*

7. Line 190: This tree (Fig 2) does not appear to be midpoint-rooted, as there should not be such extreme phylogenetic “outliers” (ST18) after rooting. Additionally, the fact that ST18, which is globally a relatively highly prevalent ST, appears to be quite genetically distinct from the rest of the cohort raises concerns about the generalizability of this cohort.

Response: *All trees presented are midpoint rooted. It is noted that the Australian cohort presented may be different to the make up in other areas of the world like the US or Europe but there still is a large diversity in the STs presented. It can be seen in Supplementary figure 2b that the study isolates are dispersed throughout the tree, indicating that they are globally relevant. Importantly, the study cohort contains STs that are both highly diverse as well as being highly clonal. Methods that categorise the relatedness of isolates and detect transmission events need to be able to cope with both types of STs and so we believe that these results are generalisable. Additional information explaining this has been added to the methods section in lines 207-231.*

8. Lines 213, 226: Though implied and alluded to in Figure 2, it needs to be delineated in the text that these methodologies (PCDR, SKA) utilize the entirety of the genome, not just the

core genome. This is an important distinction that may go unrecognized by a reader less familiar with these methods, affecting the interpretation of the results.

Response: *This has now been explicitly mentioned in the methods section for these two methods (line 273-277 and 285-287). The proportion of genome used in each method is also presented in Figure 4.*

9. For Figure 1b –the two panels should be put on the same axes so they are directly comparable. The subpanel should not be restricted to 50 differences, as it may confuse a reader and make them think the largest distance is 50.

Response: *Figure 1b is a composite of two graphs and the smaller panel is a zoomed in version of the larger graph. The zoomed in graph is meant to more easily show the 25 allelic cut off point and why it was chosen. The figure and legend has been updated to explain this more clearly.*

10. I have substantial concerns about making claims about the utility of these methods in identifying transmission when there are not any specific real-world applications that follow up on the detected potential/likely transmission events. Indeed, confirmation of spatiotemporal epidemiologic links that strengthens the assumptions is needed to draw meaningful conclusions. The whole paper hinges on this method being applicable to real-time investigations, but there is not evident application demonstrated in the manuscript.

Response: *To address this comment, we undertook a case study at a single hospital incorporating isolates collected in the first half of 2021 (Figure 6 and Lines 308-322, 464-484, 603-619). The results from this case study show that this method allows for the clear identification of wards that are likely to be involved in the transmission events identified. In addition, this paper is largely a proof of concept to show that the method chosen is fit for purpose and it is worth investing resources to ensure that the logistics of close to real-time sequencing can take place.*

11. I am not sure that the proposed workflow can function as a “one size fits all” approach. For moderately sized cohorts such as the one analyzed in the paper, this approach is reasonable, but for relatively small cohorts, such as in localized hospital outbreak settings, might it be more appropriate to use SKA as an initial clustering step instead of cgMLST, then use PCDR for the fine-resolution final step of only the most highly-related isolates. This approach would limit the computational resources necessary, and it would also enable full genome resolution of the most related isolates taking the genetic context into consideration.

Response: *This suggestion has merits, however, we wanted a method that had a broad classification step followed by a more in depth analysis. To allow for analysis of changes in population over time it is also preferable that the method be standardised. cgMLST also has similarities to the current MLST scheme, making it more easily understandable for hospital staff that will be using the results of such a pipeline. We used our proposed method on a single hospital in the case study and found it performed very well, identifying putative transmission events that could later be linked by epidemiological data.*

Reviewer #3 (Remarks to the Author):

In this paper, Higgs et al. assess different genomic approaches to identify vancomycin-resistant *Enterococcus faecium* transmission. Find below a summary of my major comments followed by all comments.

Summary of major comments:

- The authors state that “thresholds [in the literature] are not accompanied by methods to standardise the preceding steps in the analyses”. The authors should make the commands they used available to facilitate adoption of their methods.

Response: *This is a good suggestion. A file has now been included in the supplementary material that has examples of the commands used in each of the comparison methods (Supplementary Data 5).*

- It is key for interpretation of the results that the authors include further information on the study design in the Methods section. Also, they should be more explicit on how epidemiological data is used to define probable and possible transmission events. See related comments below.

Response: *The methods section has been updated to include more information on the study design and the epidemiological classification system (Line 114-142), in addition to the decision tree in Supplementary Figure 1.*

- The authors specify that PCDR (Pairwise comparison using de novo references) is “the gold standard for determining isolate relatedness”. However, this method could be compromised by inflated SNP counts due to error in the de novo assembled sequence used as reference. The authors must ensure and demonstrate that PCDR distances are not inflated by miss-assemblies.

Response: *This is a very good suggestion and we have updated our paper to incorporate this. To ensure that this self SNP inflation did not occur, we have mapped all isolate reads to their corresponding SKESA assemblies to identify any self SNPs. These SNPs were then excluded from all PCDR comparisons (Line 263-272). The PCDR distances have been updated as well as all corresponding calculations and figures in subsequent sections of the paper. Although in an ideal world we would have long read assemblies for all isolates, the current costs and time make this prohibitive in a real time detection scenario.*

- The authors should assess the impact of recombination on genetic distances, particularly given the lack of correlation between the SNP distances of different methods.

Response: *A recent study from our group published by Gorrie et al. (Lancet Microbe; [https://doi.org/10.1016/S2666-5247\(21\)00149-X](https://doi.org/10.1016/S2666-5247(21)00149-X)) performed a systematic analysis of all the key parameters for genomics-based-real-time detection and tracking of multidrug-resistant bacteria (including VREfm). One of the key findings of this paper was that removing regions of recombination “had a highly variable effect, often inflating the number of closely related pairs”, providing evidence that it is better not to mask recombination in VREfm. When accessing the relationship of very closely related isolates, as we have in this paper, we want to ensure that there are as many informative sites as possible. We also aimed to make the pairwise distance methods as similar as possible to allow for comparisons between them. PCDR and SKA could not be masked for recombination so it was decided that no methods would be masked for recombination. We had added a line in the paper to address this (Line 252-256). We have also added information from the species alignment to show that recombination screening can leave very few informative sites to use in subsequent analyses (lines 223 – 229).*

- In Supplementary Figure 11, the intra-patient diversity is rather comparable between methods (6 to 13 SNPs), but much higher for cgMLSTCA, which results in more than twice as many putative transmission links compared to other methods. How can this be explained especially when compared to cgMLSTCRCA?

Response: Thank you for pointing out the large differences in inpatient diversity between cgMLSTCA and the other methods. Supplementary Figure 5 displays all the within patient pairwise distances and how they compare across the different methods. The 90th percentile threshold was chosen to include as many same patient pairs as possible while still accounting for some outliers. Due to the more varied distribution of the cgMLSTCA distances, the 90th percentile threshold was considerably higher than the other methods. This is consistent with trends observed when comparing all the pairwise distance as shown in Supplementary Figure 10 where cgMLSTCA consistently mislabelled isolate pairs as more/less closely related than the gold standard PCDR. If pairs were consistently labelled as less closely related by cgMLSTCA then this could be adjusted for but it is the variability that is the bigger issue. An updated explanation of this has been added to the discussion (Line 562 to 571).

Comments

Abstract - line 24 “methods are not yet optimised”. It is not clear what the methods need to be optimised for.

Response: This line has been updated to “methods are not yet standardised or optimised for accuracy” (Line 31).

Methods

On the study design and data collection. The authors should include more information on the screening procedures in place at the eight hospitals. 308 positive patients seems a rather low number considering the duration of the study (15-months) and the number of participating hospitals (eight hospitals). However, this is hard to assess without knowing the exact VRE screening procedures, what wards and patient populations were targeted and the VREfm positivity obtained. At the moment it looks as if a very sparse sampling took place, which would limit the detection of transmission.

Response: The screening practices differed by hospital and would indeed affect the number of positive patients that were identified. However, the aim of this paper is not necessarily to quantify the amount of transmission occurring in each hospital but instead to compare methods that could potentially be used to detect it. Additional information regarding the specific screening practices of each of the hospitals has been added to the Supplementary Information (sup table 1) and this is referenced in the Methods section (Line 131-142).

Lines 116 to 120. In statement: “in putative transmission as defined by the “Controlling Superbugs” study (patients with an isolate determined to be genomically related to another isolate given the set threshold), and the temporospatial overlaps for each patient pair determined.” It is crucial to define how putative transmission were defined from epidemiological data in this manuscript.

Response: This has been removed from the methods section as it is confusing as highlighted by the reviewer. Although a process was used in the controlling superbugs study, it resulted in all patients with VRE having ward data collected so the methods section has been updated to reflect that to avoid confusion between processes (line 133-135).

Lines 121 to 123: “sample collection strategy, sequencing and epidemiological data collection and categorisation” need to be brought into this manuscript, as they are key for interpretation.

Response: Additional information regarding this aspect of the study has been added to the Methods section (Lines 137-142) along with detailed descriptions in the Supplementary Table 1 and Supplementary Figure 1.

Line 155. Can the authors check if Supplementary Data 1 is the one containing the “complete list of reference genomes” as specified here? Supplementary Data 1 seems to be the table containing isolate details and Supplementary Data 2 the one with the list of reference genomes.

Response: This is correct, the sup data was incorrectly numbered. The manuscript has been updated to refer to the correct files.

Line 161. In statement “These isolates are listed in supplementary data.” Please indicate what specific supplementary data file.

Response: This has been updated to Supplementary Data 2.

Lines 177 - 179. Did the authors map Illumina short reads to the consensus sequences using Snippy? If so, they need to be more explicit about the use of Illumina short reads in this statement for clarity.

Response: Thank you for this comment. This has been updated to “Following the reconcile, sequence alignment and the consensus step, the consensus sequences were polished with Illumina short reads using snippy (v4.6.0)” (Lines 199-205).

Line 195. In statement: “Only isolates from the four major STs (ST1421, ST1424, ST80 and ST203) with more than three isolates in their respective cgMLST clusters were used in the genomic comparisons (n=278).” It is not clear to this reviewer the rationale for excluding non-major STs from genomic comparisons.

Response: This was done to ensure that there were enough isolates in each group to make inferences about transmission. These are the most common STs and so are most likely to have transmission events. Increasing the numbers of STs involved in the analysis would have complicated the number of reference genomes that needed to be sequenced and increased the run time. The four major STs contain 88% of all study isolates and cover a range of diversity levels, from the highly clonal ST1421 to the very diverse ST80. As such we believe that these four STs provide an accurate representation of VRE ST dynamics.

It would be informative to specify what proportion of the genome is being used for comparisons by each method. SRA and PCDR use 100% of the genome in pairwise comparisons; but what percentage do cgMLSTCA and cgMLSTCRCA use?

Response: This information is contained in Supplementary data 2 and is itemised for each of the retrospective cgMLST clusters. It is also referenced in figure 4. An explicit reference to this data has been added to line 416-419.

PCDR (Pairwise comparison using de novo references). Lines 214 to 224. Mapping the short reads of one isolate against the de novo assembly of another could produce inflated SNPs counts due to miss-assemblies in the de novo assembly used as reference. Have the authors attempted to map the short reads of each isolate against its own SKESA assembly? In other words, what steps have the authors undertaken to make sure mis-assemblies are not producing spurious SNPs? This is particularly important as authors are using PCDR as “the gold standard for determining isolate relatedness”.

Response: This is a great suggestion and would enhance the reliability of the PCDR method. We have implemented this suggestion and adjusted the methods section and all data and figures relating to the PCDR data. PCDR distances in the paper have since been updated to reflect the new pairwise distances and information about how the correction was done, along with information about how many “self SNPs” were identified can be found in the methods (line 261-273) and supplementary figure 4.

SKA (split kmer analysis). Lines 226 to 230. More information on how the ska distance is calculated is needed here. Is it the number of k-mer mismatches?

Response: Information on SKA has been added to the methods section including a description of how SNP distances are calculated using this method. When using ska distance, a SNP is defined as “Number of split kmers found in both samples where the middle base is an A, C, G or T but differs between files” (Lines 284-287).

SNP distance threshold for transmission inference. Lines 232 - 244. Determining an appropriate SNP distance threshold for transmission analyses based on intra-patient diversity is an established approach. But first, the clonality of isolates from the same patient must be confirmed (to avoid comparing distant isolates from different acquisition events). Did the authors check that isolates from different cgMLST in the same patient were much more distant (in terms of number of SNPs) than isolates from the same cgMLST?

Response: Thank you, this is an important point. cgMLST clusters and SNPs distances are somewhat comparable, ie if two isolates have more than 25 allelic differences by cgMLST then they are guaranteed to have at least 25 SNPs between them. However, due to the differences between the way that allelic differences and SNP differences are calculated we cannot always be sure that they are proportional. There were 8 patients that had isolates from different cgMLST clusters comprising 10 isolate pair combinations. Using SKA (a method shown to be highly accurate), these isolate pairs were shown to have a pairwise distance ranging from 396 to 3176 SNPs. This is considerably more than within patient isolate pairs from the same cgMLST cluster which never exceeded 34 SNPs. This information has been added to the methods section at lines 295-298.

In Supplementary Figure 11, the intra-patient diversity is rather comparable between methods (6 to 13 SNPs), but much higher for cgMLSTCA; how can this be explained?

Response: Supplementary Figure 5 displays all the within patient pairwise distances and how they compare across the different methods. The 90th percentile threshold was chosen to include as many same patient pairs as possible while still accounting for some outliers. Due to the more varied distribution of the cgMLSTCA distances, the 90th percentile threshold was considerably higher than the other methods. This is consistent with trends observed when comparing all the pairwise distances as shown in Supplementary Figure 10 where cgMLSTCA consistently mislabelled isolate pairs as more/less closely related than the gold standard PCDR. If pairs were consistently labelled as less closely related by cgMLSTCA then this could be adjusted for but it is the variability that is the bigger issue. An updated explanation of this has been added to the discussion (Line 559 to 571).

Results

Figure 1. It would be advisable to root the phylogenetic tree. The authors should assign E. faecium isolates to clade A or B. Most likely, the largest clade in Supplementary Figure 10 corresponds to clade A, as this is the most commonly isolated clade among hospital isolates, but this needs to be confirmed with labelled contextual isolates. The two outliers in red at the top of this figure might correspond to clade B or basal clade A isolates. If that's the

case, these isolates can be used as an outgroup to root the clade A.

Response: *The phylogenetic tree is not being used to make any conclusions about the origin of isolates or how groups related to one another. Instead, it is being used to simply show the diversity of the study population and the lack of correlation between MLST and the phylogenetic clustering of isolates. As suggested, there is interest in clade A vs B and we have added a phylogenetic tree that contains other isolates for a global context (Supplementary figure 2b). This tree indicates that most of our isolates belong to clade A1. Three highly divergent study isolates that were shown to cluster with clade A2 and B were excluded from the species tree shown in the main paper. This is detailed in lines (208 –221).*

In this section, the authors compare the agreement between MLST and cgMLST clusters with the population structure, as defined by the phylogenetic tree. Recombination needs to be detected and removed prior to building the phylogenetic tree (using tools such as Gubbins), to obtain a robust phylogenetic tree.

Response: *When building a tree from such a diverse data set such as the one in the figure, use of recombination filtering is not recommended and usually results in the loss of a very large amount of genetic information. This is further addressed in the comment below to the next author query.*

MLST is known not be a robust genotypic scheme for *E. faecium*, the authors should cite studies reporting this. Indeed, it is expected for some major STs to be polyphyletic, i.e. they do not fall within single monophyletic clades. Most cgMLST clusters seem to be monophyletic, which is good, with the exception of cgMLST cluster 6 and 17. It would be good to check if they become monophyletic once a recombination-free and rooted phylogenetic tree is used.

Response: *A recent study from our group published by Gorrie et al. (Lancet Microbe) performed a systematic analysis of all of the key parameters for genomics-based-real-time detection and tracking of multidrug-resistant bacteria (including VRE). One of the key findings of this paper was that removing regions of recombination “had a highly variable effect, often inflating the number of closely related pairs”. We wanted the phylogenetic trees to reflect the pairwise distances used elsewhere in the paper and so screening for recombination was not performed. For the species tree, we also did not use recombination masking as due to the diversity of the isolates, post Gubbins the core genome alignment was only 39 SNP sites. Building a species tree on an alignment this small would not be appropriate. This information has been added to the methods section better articulate why recombination masking was not used (lines 223-229).*

The use of a 25 cgMLST SNP threshold to define clusters seems reasonable, given the distribution of cgMLST SNP distances shown in Figure 1b.

Response: *No changes needed in paper.*

Line 318 - 319. In statement “pairwise SNP distance ≤ 50 ”. Please indicate what type of SNP distances, of the different ones calculated in this work, are referred to here.

Response: *This has been updated to now read “(PCDR pairwise SNP distance ≤ 50)”.* (Line 403).

Section “For very closely related isolate pairs, direct pairwise comparison tools should be used”. Lines 302 - 332. I found unexpected that core-genome SNP distances do not correlate with whole-genome distances (i.e. PCDR distances). As explained in a point above, the authors must ensure and demonstrate that PCDR distances are not inflated by

miss-assemblies. It would be interesting to explore if the lack of correlation is due to recombination, which again, is expected to inflate SNP counts. As recombination is expectedly lower, or absent, among highly related strains that diverged recently, it would be interesting to test for correlation separately among highly and distantly related pairs.

Response: *As per suggestion from the reviewer, the PCDR distances have now been updated to account for any self SNPs. These have been removed from all PCDR comparisons and the thresholds updated accordingly. Supplementary Figure 4 shows the distribution of the number of self SNPs identified. The methods have been updated to reflect the changes (Line 261-273).*

Lines 337 to 344. As pointed above, in Supplementary Figure 11, the intra-patient diversity is rather comparable between methods (6 to 13 SNPs), but much higher for cgMLSTCA, which results in more than twice as many putative transmission links compared to other methods. How can this be explained especially when compared to cgMLSTCRCA?

Response: *This is largely due to the diversity of the reference. The isolates within each cluster are the same. Having a more diverse reference means that there are more variant sites to include in the core alignment. You can see from Supplementary Figure 5 that the majority of cgMLSTCA same patient pairwise distances are very small and comparable to the other methods, it is just that they are more broadly distributed than other methods. A similar trend can be seen in Supplementary Figure 10.*

Lines 367 to 375.

Response: *No comment to respond to.*

372 to 375. The definitions of epidemiological evidence (i.e. probable and possible transmission likelihood) need to be included in the methods, as it is key for interpretation of these results.

Response: *These definitions have now been included in the methods section (Line 136-142), in addition to the decision tree in the supplementary data.*

Figure 4 and Supplementary Figure 9. In addition to reporting the proportion of isolate pairs that are genomically linked under the different methods/thresholds, it would be relevant and more intuitive to report the proportion of patients that are genomically linked too.

Response: *This is a good suggestion and has been updated now be included in Figure 4. For Sup Figure 9, the figure would be virtually the same as this figure already incorporates a same patient classification and so has not been changed.*

Please make sure the numbering of supplementary figures is consistent with the order they appear in the manuscript.

Response: *The numbering of the supplementary figures has been updated to reflect the order in which they appear in the manuscript.*

REVIEWERS' COMMENTS

Reviewer #1 (Remarks to the Author):

I am fine with your responses

Reviewer #2 (Remarks to the Author):

I believe the authors have made an important effort to address the concerns of the reviewers. In particular, addressing the real-life application of the methodology proposed. Although I would have liked to see a more robust validation, the test case they present is an initial step in the right direction, although I still believe the authors have room on improving this effort. Although the methodological issues do not increase the novelty, the information provided now offers much more confidence that this approach may be useful in clinical settings and, hopefully, can be validated in other geographical locations.

Minor comments:

Lines 284-286: The phrasing of this added sentence is a little confusing-- it depicts SKA as a lesser method of analysis as compared to PCDR. Consider rephrasing to, "In contrast to a traditional core genome alignment, SKA compares isolate pairs based on all genomic content contained in the reads file, which is equivalent to a whole genome comparison."

Line 614: "were" should be "where"

Figure 6: "epidemiological" is misspelled

Reviewer #3 (Remarks to the Author):

The latest version of the manuscript and rebuttal show that the authors have taken into consideration all my comments and incorporated new analyses, changes or clarifications when needed. I have thus a few minor remaining comments:

- The abstract needs to be improved to include all major conclusions. For instance, the authors do not include their recommendation of using cgMLST as a clustering method for Efm. Also SKA should be named in the abstract when referring to the "kmer-based approach". The fact that "cgMLSTCRCA, PCDR and SKA all had a similarly high level of genomic putative transmission links that were supported by epidemiological evidence" (~80%) is also important, but absent in the Abstract.

- In lines 110-112. In the concluding statement of the Introduction "Here, we show the benefits of using", the authors may want to use a more strong statement such as "Here, we show the superiority of using".

- Correct instances of "enterococcus faecium" with "Enterococcus faecium"

REVIEWERS' COMMENTS

Reviewer #1 (Remarks to the Author):

I am fine with your responses

Response: No comment to respond to.

Reviewer #2 (Remarks to the Author):

I believe the authors have made an important effort to address the concerns of the reviewers. In particular, addressing the real-life application of the methodology proposed. Although I would have liked to see a more robust validation, the test case they present is an initial step in the right direction, although I still believe the authors have room on improving this effort. Although the methodological issues do not increase the novelty, the information provided now offers much more confidence that this approach maybe useful in clinical settings and, hopefully, can be validated in other geographical locations.

Minor comments:

Lines 284-286: The phrasing of this added sentence is a little confusing-- it depicts SKA as a lesser method of analysis as compared to PCDR. Consider rephrasing to, "In contrast to a traditional core genome alignment, SKA compares isolate pairs based on all genomic content contained in the reads file, which is equivalent to a whole genome comparison."

Response: This is a good suggestion and lines 284-286 has been changes to the phrasing suggested.

Line 614: "were" should be "where"

Response: Main text has been changed to where.

Figure 6: "epidemiological" is misspelled

Response: Figure legend has been changed.

Reviewer #3 (Remarks to the Author):

The latest version of the manuscript and rebuttal show that the authors have taken into consideration all my comments and incorporated new analyses, changes or clarifications when needed. I have thus a few minor remaining comments:

- The abstract needs to be improved to include all major conclusions. For instance, the authors do not include their recommendation of using cgMLST as a clustering method for Efm. Also SKA should be named in the abstract when referring to the "kmer-based approach". The fact that "cgMLSTCRCA, PCDR and SKA all had a similarly high level of genomic putative transmission links that were supported by epidemiological evidence" (~80%) is also important, but absent in the Abstract.

Response: This is a good suggestions and SKA has now been directly referred to in the abstract, as has the cgMLST clustering approach. Due to the word limit constraints, we were not able to include the information about the comparison of epidemiological evidence as it would involve introducing all of the other genomic methods used.

- In lines 110-112. In the concluding statement of the Introduction "Here, we show the benefits of using", the authors may want to use a more strong statement such as "Here, we show the superiority of using".

Response: Line 111 has been updated to read “Here, we show the superiority of using”.

- Correct instances of “enterococcus faecium” with “Enterococcus faecium”

Response: The main text has been updated to correct all instances.